# SARS-CoV-2–host proteome interactions for antiviral drug discovery

Xiaonan Liu[1,2] iD, Sini Huuskonen[1,2] iD, Tuomo Laitinen[3,†] iD, Taras Redchuk[1,2,†] iD,
Mariia Bogacheva[2,4,5,†] iD, Kari Salokas[1,2] iD, Ina Pöhner[3] iD, Tiina Öhman[1,2] iD,
Arun Kumar Tonduru[3] iD, Antti Hassinen[2,4] iD, Lisa Gawriyski[1,2] iD, Salla Keskitalo[1,2] iD,
Maria K Vartiainen[1,2] iD, Vilja Pietiäinen[2,4] iD, Antti Poso[3,6] iD & Markku Varjosalo[1,2,*] iD

## Abstract

Treatment options for COVID-19, caused by SARS-CoV-2, remain limited. Understanding viral pathogenesis at the molecular level is critical to develop effective therapy. Some recent studies have explored SARS-CoV-2–host interactomes and provided great resources for understanding viral replication. However, host proteins that functionally associate with SARS-CoV-2 are localized in the corresponding subnetwork within the comprehensive human interactome. Therefore, constructing a downstream network including all potential viral receptors, host cell proteases, and cofactors is necessary and should be used as an additional criterion for the validation of critical host machineries used for viral processing. This study applied both affinity purification mass spectrometry (AP-MS) and the complementary proximity-based labeling MS method (BioID-MS) on 29 viral ORFs and 18 host proteins with potential roles in viral replication to map the interactions relevant to viral processing. The analysis yields a list of 693 hub proteins sharing interactions with both viral baits and host baits and revealed their biological significance for SARS-CoV-2. Those hub proteins then served as a rational resource for drug repurposing via a virtual screening approach. The overall process resulted in the suggested repurposing of 59 compounds for 15 protein targets. Furthermore, antiviral effects of some candidate drugs were observed *in vitro* validation using image-based drug screen with infectious SARS-CoV-2. In addition, our results suggest that the antiviral activity of methotrexate could be associated with its inhibitory effect on specific protein–protein interactions.

**Keywords** drug discovery; mass spectrometry; proteomics; SARS-CoV-2; virus–host interactions

**Subject Categories** Microbiology, Virology & Host Pathogen Interaction; Pharmacology & Drug Discovery; Proteomics

**Mol Syst Biol. (2021) 17: e10396**

## Introduction

The ongoing global coronavirus disease 2019 (COVID-19) pandemic is caused by severe acute respiratory syndrome (SARS) coronavirus 2 (SARS-CoV-2) (Wu *et al*, 2020). Treatment options for COVID-19 are limited, and consequently, coronavirus infections are overwhelming national healthcare systems. The R&D activity to develop a vaccine or drug against COVID-19 is being fast-tracked globally. By July 2021, 32 vaccines had reached phase-three clinical trials, and 11 were approved by at least one country (Dai & Gao, 2020; Creech *et al*, 2021). Although vaccines are the primary means to prevent COVID-19, antiviral drugs would significantly reduce the disease burden for the early treatment of COVID-19, and long COVID (Schmidt, 2021) suppression. Remdesivir (veklury), an experimental drug that was originally investigated as a potent inhibitor of Ebola virus (EBOV) (Warren *et al*, 2016), was the first drug approved by the US Food and Drug Administration (FDA) for the treatment of COVID-19 in October 2020 (Rubin *et al*, 2020). However, the World Health Organization (WHO) panel advised physicians against using remdesivir based on a review of several large clinical trials (preprint: WHO 2020; Harrington *et al*, 2021). Despite the controversy over whether remdesivir can reduce the duration of COVID-19, it is obvious that fully effective drugs for the prevention or treatment of SARS-CoV-2 are currently not available.

Antiviral drugs mainly include direct virus-targeting and host-targeting antiviral drugs. Virus-targeting drugs often inhibit viral polymerases and proteases, while host-targeting drugs aim to

1 Institute of Biotechnology, University of Helsinki, Helsinki, Finland
2 Helsinki Institute of Life Science, University of Helsinki, Helsinki, Finland
3 School of Pharmacy, University of Eastern Finland, Kuopio, Finland
4 Institute for Molecular Medicine Finland, University of Helsinki, Helsinki, Finland
5 Department of Virology, University of Helsinki, Helsinki, Finland
6 Department of Internal Medicine VIII, University Hospital Tübingen, Tübingen, Germany
*Corresponding author. Tel: +358 2941 59413; E-mail: markku.varjosalo@helsinki.fi
†These authors contributed equally to this work

disrupt the virus–host protein interactions that are essential for viral replication. Due to the high viral evolutionary rates, resistance to virus-targeting drugs can occur and lead to treatment failure, especially for infections caused by RNA viruses (Heaton, 2019). In contrast, such effects can be avoided by using host-targeting drugs because of the low evolutionary divergence of host proteins. Therefore, it is necessary to construct a comprehensive virus–host proteome interaction atlas that can be used to identify the cellular functions that are mandatory for viral processing and, in turn, to develop effective therapeutic strategies against SARS-CoV-2 and new emergent strains.

Four recent proteomic studies (Gordon *et al*, 2020b; preprint: Laurent *et al*, 2020; preprint: Samavarchi-Tehrani *et al*, 2020; preprint: Stukalov *et al*, 2020) uncovered extensive SARS-CoV-2–host protein–protein interaction (PPI) networks. These studies used known SARS-CoV-2 virus open-reading frames (ORFs) as baits to identify interacting proteins by affinity purification or proximity purification combined with mass spectrometry (AP-MS or BioID-MS) in HEK293 or A549 cells. AP-MS (Varjosalo *et al*, 2013) is suitable for the study of virus–host multiprotein complexes, while BioID-MS (Roux *et al*, 2012) has become a complementary method to capture transient interactions that frequently occur throughout viral infection progression. Together, these data provide biochemical insights into how the SARS-CoV-2 hijacks host cells.

However, in identifying potential drug targets, studies focusing on viral bait protein interactions are inherently biased, as they neglect to incorporate the cellular context of the host. For example, angiotensin-converting enzyme 2 (ACE2) is a known receptor of SARS-CoV-2 (Gheblawi *et al*, 2020). Because of its low expression level in most experimental cell lines (Hikmet *et al*, 2020), none of the abovementioned studies were able to detect the interaction between the spike protein and ACE2 (Wan *et al*, 2020). Furthermore, the interactions of viral proteins with host proteins have cascading effects on the host interactome, where certain essential proteins necessary for the viral replication cycle are indirectly affected. Therefore, consideration of the downstream host protein interactome is necessary and should be used as an additional criterion for selecting potential targets for therapeutic intervention.

As a contribution to this effort, we adopted the Multiple Approaches Combined (MAC)-tag system (Liu *et al*, 2018, 2020), enabling both AP-MS and BioID-MS analysis with a single construct to perform a comprehensive analysis of all 29 SARS-CoV-2 and 18 host proteins, which include cell surface receptors, proteases, restriction factors, replication factors, and trafficking factors, with known roles in viral infection processes. We first generated a virus–host interactome and compared this PPI network with the results of other studies to address functionally conserved protein interactions. Subsequently, we characterized 693 hub proteins that connect viral baits and host baits via a dense network to reveal critical pathways in the host used for viral replication. Selected hub proteins were then used to propose drug repurposing candidates by a virtual screening approach. Our analysis finally prioritized 59 promising drug candidates for use against SARS-CoV-2. Furthermore, ten candidate drugs were further validated using image-based drug screen with infectious SARS-CoV-2, six of them demonstrating antiviral effects.

# Results

## Global analysis of SARS-CoV-2 ORFs and its host protein interactome

SARS-CoV-2 is a positive-sense single-stranded (+)ssRNA virus, ~29.9 kb in size, and it contains 14 ORFs encoding 29 proteins (Chan *et al*, 2020). The first ORF (ORF1a/b) produces two polypeptides, namely, pp1a and pp1ab, which are further processed into 16 nonstructural proteins (NSPs). ORFs 2 to 14 encode four main structural proteins, namely, the spike (S, ORF 2), envelope (E, ORF 4), membrane(M, ORF 5) and nucleocapsid (N, ORF 9), and nine additional accessory factors (ORF 3a, 3b, 6, 7a, 7b, 8a, 8b, 9b, and 9c) (Fig 1A). The viral S, M, and E proteins are embedded in the lipid membrane on the virion surface. The N protein interacts with viral RNA in the core of the virion (Fig 1A). SARS-CoV-2 utilizes ACE2 and transmembrane serine protease 2 (TMPRSS2) as a prime receptor and a critical protease, respectively, to enter target cells (Fig 1A) (Hoffmann *et al*, 2020). Alternative receptors have also been reported, including CD147 (BSG) (Wang *et al*, 2020), neuropilin-1 (NPR1) (Cantuti-Castelvetri *et al*, 2020; Daly *et al*, 2020), transferrin receptor (TFRC) (preprint: Tang *et al*, 2020), and C-type lectin domain family 4 member D/E (CLEC4D/CLEC4E) (preprint: Singh *et al*, 2020; Yi & Chuanxin, 2020). In addition to TMPRSS2, several other cellular proteases work as alternative priming factors, including TMPRSS4 (Zang *et al*, 2020), TMPRSS11A/B (Zhang, Zhang, *et al*, 2020), FURIN (Xia *et al*, 2020), and cathepsin B, L, and S (CTSB, CTSL, and CTSS) (Vieira Braga *et al*, 2019). Furthermore, a genome-scale loss-of-function screen discovered that DNA topoisomerase III beta (TOP3B) is required for efficient replication of a diverse group of (+)ssRNA viruses, including SARS-CoV-2 (Fig 1A) (Prasanth *et al*, 2020). Some membrane proteins that are known as receptors or regulators of other coronaviruses, such as dipeptidyl peptidase 4 (DPP4) (receptor of hCoV-EMC) (Raj *et al*, 2013), aminopeptidase N (ANPEP) (receptor of coronavirus TGEV) (Delmas *et al*, 1992), and interferon-inducible transmembrane proteins (IFITM1 and IFITM3) (Huang, Bailey, *et al*, 2011; Shi *et al*, 2021), may not directly bind to SARS-CoV-2 proteins, but can enhance viral entry (Fig 1A) (Li *et al*, 2020).

We aimed to identify host proteins associated with SARS-CoV-2 proteins systematically using both AP-MS and BioID-MS. To achieve this, we cloned the 29 genes corresponding to SARS-CoV-2 proteins (Fig 1A (left) and Dataset EV1) and 18 host proteins (Fig 1A (right) and Dataset EV1) that are functionally relevant for SARS-CoV-2 entry and replication. Each clone was fused to the MAC-tag system (consisting of both StrepIII and BirA* tags) to generate isogenic tetracycline-inducible HEK293 cell lines (Fig 1B). The expression of MAC-tagged host proteins was confirmed by MS analysis. Considering the inefficiency of shotgun proteomics in resolving very small proteins (≤ 50 amino acids), MAC-tagged viral protein expression was confirmed by Western blotting with HA antibodies (Appendix Fig S1). All viral ORFs except NSP3 and NSP6 were successfully detected by Western blotting (Appendix Fig S1). The expression of viral peptides of NSP3 and NSP6 was then confirmed by MS analysis.

Viral infection requires the host immune response to create the cellular environment for viral protein processing. Here, a single ORF was expressed in each corresponding cell line, and some

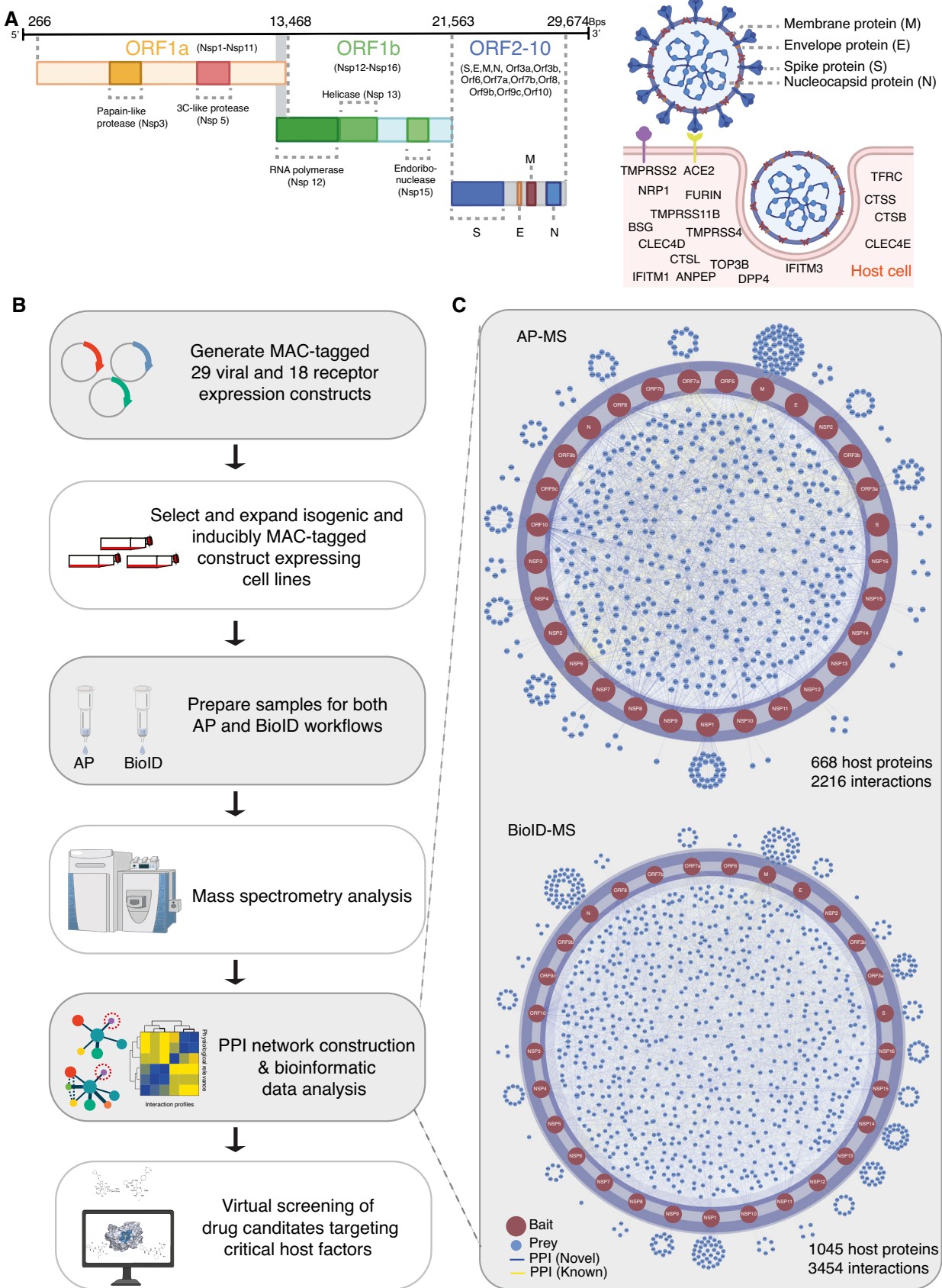

**Figure 1.**

Figure 1. Workflow for identification of SARS-CoV-2 virus–host PPIs.

A SARS-CoV-2 genome annotation. Two-thirds of the viral RNA, mainly located in the first open-reading frame (ORF 1a/b), encodes 16 nonstructural proteins (NSPs). The rest of the viral genome encodes four main structural proteins, namely the spike (S), envelope (E), nucleocapsid (N), membrane (M) proteins, and several accessory factors. SARS-CoV-2 enters the cell primarily via binding to ACE2, followed by its priming by TMPRSS2. In addition to ACE2 and TMPRSS2, other potential SARS-CoV-2 receptors, proteases, and cofactors for infection are indicated.
B Experimental workflow for the establishment of a SARS-CoV-2–host interactome to identify potential drug targets and treatment.
C Virus–host protein interaction networks were constructed by AP-MS (top) and BioID-MS (bottom). Only protein–protein interactions (PPIs) with a Bayesian false discovery rate (BFDR) ≤ 0.01 (assessed by SAINTexpress) are shown.

cooperative viral interactions were possibly missed. We therefore monitored the immune response of the host cell lines to check whether these stable cell lines reflect the cellular context of viral infection. In total, nine viral ORF-expressing cell lines were randomly selected for transcriptomic profiling (Appendix Fig S2A and Dataset EV2). Five out of nine cell lines showed upregulated expression of FURIN (Appendix Fig S2A and Dataset EV2), which can promote SARS-CoV-2 infectivity and cell-to-cell spread (Papa et al, 2021). In contrast, viral ORF expression suppressed the expression of human leukocyte antigen (HLA)-encoded class II genes (DQA, DPB1, DPA1, DMB) and interferon-inducible transmembrane genes (IFITM1, IFITM2) (Appendix Fig S2A and B), which were previously demonstrated to facilitate matured coronavirus infection in vitro (Josset et al, 2013; Shi et al, 2021). Overall, the stable viral ORF-expressing cell lines created a cellular context mimicking viral infection.

All the stable cell lines were used parallelly for both AP and BioID purification to achieve a completed protein interactome, altogether, totaling 366 samples for MS characterization (Dataset EV3). This analysis yielded 5,670 (2,216 for AP-MS; 3454 for BioID-MS) high-confidence interactions (HCIs) for 29 viral bait proteins using both AP-MS and BioID-MS methods (Fig 1C and Dataset EV4). For the host–receptor/cofactor interaction network, we identified 5,351 (2,224 for AP-MS; 3,127 for BioID-MS) HCIs connecting 18 human bait proteins localized in different cellular compartments (Appendix Fig S3 and Dataset EV4). Notably, many viral baits (M, NSP10, NSP16, NSP6, NSP7, NSP9, ORF10, ORF3a, ORF3b, and ORF7a) were detected to interact with TFRC, and few viral baits (N, NSP13, and NSP14) interacted with TOP3B. In general, BioID-MS tends to provide more high-confidence interacting proteins (viral bait-prey pairs: 3,454; host bait-prey pairs: 3,127) than AP-MS (viral bait-prey pairs: 2,216; host bait-prey pairs: 2,224) (Dataset EV4), indicating that BioID-MS can capture highly transient and close-proximity interactions.

## Proteome interaction data validation

To verify host–receptor PPI data, we integrated and compared protein interaction data from seven major databases including Human Cell Map (Go et al, 2021), BioPlex (Huttlin et al, 2021), BioGRID (Oughtred et al, 2021), HuRI (Luck et al, 2020), PINA2 (Cowley et al, 2012), STRING (Szklarczyk et al, 2019), and IntAct (Kerrien et al, 2012). In total, 2,465 known interactions related to 18 host bait proteins were retrieved (Fig EV1A), and more than half of them were seen in only one database highlighting different evidence channels (colocalization, coexpression, literature mining, experimental evidence) of different databases (Fig EV1B). While some baits are well studied with many known interactions (e.g., BSG, TFRC, TOP3B), most have only a few reported interactions (e.g.,

ACE2, CTSS, CLEC4D) (Fig EV1C), emphasizing the need for a systematic study. Overall, 93 interactions among the 4,362 unique HCIs (~ 2.1%) were previously reported for the host protein interaction network (Fig EV1D and Dataset EV4).

Since the tissue distribution and expression of the appropriate receptors determine the tropism of the viral infection, we further investigated receptor bait protein expression in various human organs. Information was obtained from the Human Protein Atlas (Uhlén et al, 2015), and 13 of the 18 host baits were detected at least once in 45 human organs (Fig EV1E). Although their expression patterns vary greatly in different tissues, almost all detected bait proteins are expressed in the kidney (Fig EV1E), further supporting the application of the HEK293 cell model. The renal tropism of these baits is a potential explanation for common kidney injury being observed in patients with COVID-19 (Pei et al, 2020).

To validate the virus–host PPIs, we compared our study with four published/preprint studies (Gordon et al, 2020b; preprint: Laurent et al, 2020; preprint: Samavarchi-Tehrani et al, 2020; preprint: Stukalov et al, 2020) that utilized either AP-MS or BioID-MS to map SARS-CoV-2 viral ORFs interacting with host proteins in HEK293 and A549 cells (Dataset EV5). Coverage in terms of the number of bait–prey interaction pairs and the number of unique prey proteins was analyzed. Although each viral protein is expected to have specific interactions accounting for host specificity and pathogenesis, a large portion of host prey proteins are most likely shared across multiple viral baits, considering that the viral proteins are processed by the same replication machinery. Using AP-MS, 103 interactions were characterized by more than one study, and five among those were reported in all independent studies (Fig EV2A). The low overlap between independent AP-MS studies suggests that viral proteins are unlikely to form very stable complexes with host proteins and that virus–host protein interactions are more transient. With BioID-MS, over 10 times more interactions were detected across at least two studies, and 94 PPIs were highly conserved in all studies (Fig EV2B and C). Moreover, prey proteins obtained by AP-MS were mainly detected by one bait protein, while approximately half of the prey proteins were observed to interact with more than one viral bait using the BioID-MS approach (Fig EV2D–G). This implies that viral ORFs may appear in the same subcellular region; therefore, similar proximal proteins were detected. Alternatively, it may also suggest that the virus targets the same host factor in redundant ways, and further investigation is needed.

Despite the valuable insights provided by the protein interaction network, our interactome data could contain noise and can include some false-positive interactions without biological relevance. To assess the accuracy of the interaction information we provided, 95 protein interaction pairs from each dataset were selected and

validated via reverse co-immunoprecipitation (co-IP) using affinity matrix binding prey proteins to pull down bait proteins (Dataset EV6). The positivity ratios were 67% (Fig EV3A) and 77% (Fig EV3B) for the viral interaction dataset and host–receptor interaction dataset, respectively. This is higher than the true-positive ratio of common protein databases (20–40%) (Kuchaiev *et al*, 2009; Kotlyar *et al*, 2019). Moreover, different baits targeting the same prey protein were also validated (Fig EV3 and Dataset EV6). For example, we assessed 12 interaction pairs of viral baits interacting with ATP-dependent 6-phosphofructokinase (PFKP), and six out of 12 were confirmed as positive by Co-IP (Fig EV3 and Dataset EV6).

We further conducted a functional enrichment analysis on the viral protein-associated and host–receptor-associated proteins. The interaction partners of the structural proteins of SARS-CoV-2 (S, M, and E), as well as several other viral proteins, were significantly enriched in the calnexin/calreticulin cycle (Appendix Fig S4), and calreticulin family domain (Appendix Fig S5), indicating that these proteins bud into the lumen of the ER-Golgi intermediate compartment (ERGIC) to form mature virions (Harrison *et al*, 2020). The N protein interactome was significantly enriched in the regulation of stress granule assembly (Appendix Fig S6), which was confirmed in a previous study (preprint: Samavarchi-Tehrani *et al*, 2020). Moreover, several viral protein interactomes were closely related to meiotic movement and cytokinesis (Appendix Fig S6), indicating that SARS-CoV-2 infection could perturb the cell cycle (Bouhaddou *et al*, 2020; Tutuncuoglu *et al*, 2020).

The interactomes of several host receptors (e.g., TFRC, NRP1, ACE2, FURIN) showed enrichment of Nef-mediated downregulation of CD4 and CD8 (Appendix Fig S7). It can be speculated that these receptors regulate the immune response during infection. Predictably, the same set of host receptors was involved in the clathrin-related endocytosis process (Appendix Fig S8), which is activated during the internalization of a large number of viruses (e.g., influenza virus, SARS coronavirus, Hepatitis B virus) (Brodsky *et al*, 1992; Veiga & Cossart, 2006; Maksymowicz *et al*, 2020). Moreover, domain-based enrichment analysis showed that adaptin domains were overrepresented in the interactomes of several receptors (TOP3B, CLEC4D, CLEC4E, TMPRSS2, IFITM1, IFITM3), indicating their roles in cellular trafficking (Appendix Fig S9).

Even though abovementioned high-throughput studies were carried out by experienced proteomic researchers, each dataset contains a substantial level of unique interactions inherent to the cell lines used for the study, the workflow used for protein purification, and the data validation strategy. For these reasons, profiling the interactome of SARS-CoV-2 with different approaches allows the recovery of distinct interactions and can be considered a work in progress.

### Subcellular localization of viral bait proteins

The subcellular distributions of viral proteins are associated with unique biological functions. To systematically study the subcellular localization of SARS-CoV-2 proteins and to evaluate the potential morphological changes caused by the expression of these proteins, affinity-tagged viral ORFs were transfected into U2-OS cells and detected by immunofluorescence. Considering that coronavirus infection can disrupt host cytoskeleton homeostasis which is tightly connected to pathological processes (Wen *et al*, 2020), we decided to use phalloidin to evaluate general cell morphology and cytoskeleton integrity (Fig 2). Except for ORF9b, for which we failed to image transfected cells, all other 28 viral proteins were successfully expressed in U2-OS cells. The results indicated a diversity of subcellular locations of viral proteins in cells. These included ORF7a, M, ORF6, and ORF3b, which were predominantly visible in the Golgi apparatus, and ORF10, NSP4, and S, which were localized in the ER. In addition, a comparison of the results with previous publications (Gordon *et al*, 2020a; Zhang, Cruz-cosme, *et al*, 2020; Lee *et al*, 2021) showed that in most cases there was no obvious difference observed in the subcellular localization of individual ORFs in several transfected cell lines (Dataset EV7). To further characterize the potential compartment specificity of viral ORFs, we employed our developed MS-microscopy system (Liu *et al*, 2020), which uses a quantitative interactome profile to map the cellular distribution of the bait protein (Fig EV4A and Dataset EV7). This revealed two baits (NSP16 and ORF3a) that were associated with endosomes, five baits (S, E, ORF7b, ORF8, and ORF10) that showed an ER distribution (Fig EV4A), and four baits (M, ORF6, ORF7a, and NSP10) that were related to the Golgi apparatus. ORF9b was mainly localized to the mitochondrion (Fig EV4A). Although coronavirus replication occurs in the cytoplasm of infected host cells, 18 bait proteins (including NSP1, NSP5, NSP6, NSP7, NSP9, NSP14, NSP16, and N) had nucleus-related PPI networks. Overall, our results are consistent with previous systemic imaging studies of the subcellular localization of these proteins (Gordon *et al*, 2020a; Zhang, Cruz-cosme, *et al*, 2020) and confirm that our protein interaction data correctly reflect the subcellular molecular context (Dataset EV7).

### NSP3 modulates host actin for F-actin ring formation

Although the MS-microscopy system detected several bait proteins with interactions related to actin filament (NSP3, NSP10, and N) or intermediate filaments (NSP2, NSP4, NSP8, NSP9, NSP12, NSP13, NSP14, and ORF3b) (Fig EV4A), we did not observe any obvious defects in actin cytoskeletal organization in most of the transfected cells. In cells expressing most ORFs, phalloidin staining showed an actin meshwork that extended throughout the cytoplasm. (Fig 2). However, a unique actin filament structure that was associated with and extended into the nucleus was observed in NSP3-transfected cells (Figs 2 and EV4A). In most cells, this structure consisted of a rim of actin filaments delineating the nuclear envelope, reminiscent of the perinuclear actin cage observed in mechanically stimulated cells (Shao *et al*, 2015). Confocal microscopy revealed a meshwork of actin filaments inside the cell nucleus (Fig EV4A). These structures were striking, since nuclear actin filaments are usually detected only under specific stimuli or conditions, such as serum stimulation, cell adhesion, DNA damage response, or the early G1 phase of the cell cycle (Ulferts *et al*, 2020). Qualitatively, the nuclear actin filament pattern detected here appeared different from these previously reported structures, with a very dense meshwork of phalloidin staining. Interestingly, some viruses that replicate in the nucleus, including alpha-herpesvirus and baculovirus, have been reported to induce nuclear actin polymerization, which is essential for virus production (Hepp *et al*, 2018; Ohkawa & Welch, 2018). However, SARS-CoV-2 replicates in the cytoplasm, and NSP3 did not localize to the nucleus (Fig 2). It therefore likely exerts its effects from the cytoplasmic side of the nuclear envelope. Both the

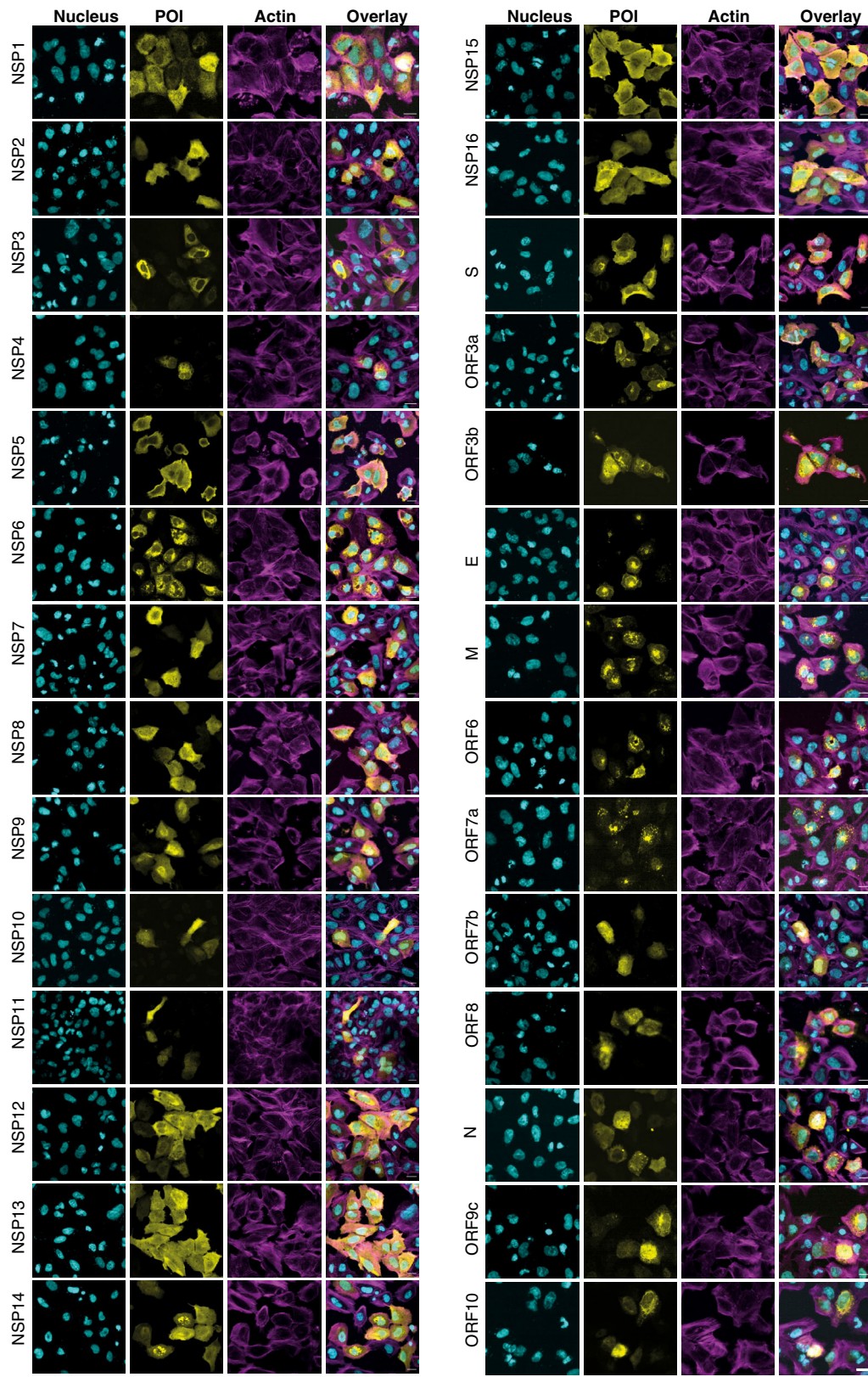

**Figure 2. Subcellular localization of the MAC-tagged viral protein.**

Cells were fixed, and the viral bait fused with the V5-tag was visualized by immunofluorescence staining using Alexa Fluor 488 labeled anti-V5 immunostaining (yellow), actin with Alexa Fluor 594-labeled phalloidin (magenta), and nuclei were stained with DAPI (cyan) (Scale bar: 10 μm).

formation of the perinuclear actin ring and nuclear actin polymerization downstream of G protein-coupled receptors, e.g., upon serum stimulation, are dependent on $Ca^{2+}$ signaling and the formin INF2 (Shao *et al*, 2015; Wang *et al*, 2019). Recent cryo-electron microscopy studies have suggested a central role for NSP3 in the formation of molecular pores through the double-membrane vesicles (Wolff, Limpens, *et al*, 2020). This pore formation activity may result in the release of $Ca^{2+}$ from the ER and thus trigger nucleus-associated actin polymerization in NSP3-expressing cells. The assembled structure around the nucleus may function as a barrier to protect host cell genome integrity until cellular homeostasis is re-established.

On the other hand, since NSP3 interacts with several myosin proteins and actin (Fig EV4B and Dataset EV4), it could also directly influence actin dynamics at least on the outer nuclear membrane. Alternatively, NSP3 could control actin reorganization by regulating relevant phosphorylation events (Fig EV4C), as has been suggested previously for SARS-CoV (Surjit *et al*, 2004). Indeed, quantification of phosphorylation changes in the NSP3-expressing cell line compared with the control revealed upregulated phosphorylation events that were involved in four signaling pathways: AMPK signaling, insulin signaling, regulation of the actin cytoskeleton, and the mTOR signaling pathway (Fig EV4D and Dataset EV8).

Previous studies have reported observations of actin reorganization by viruses; however, to our knowledge, perinuclear actin polymerization is revealed here for the first time. The detailed mechanism underlying the formation of this perinuclear actin ring structure in NSP3-expressing cells needs further clarification.

### Bait self-organization reveals distinct biological functional groups

Viral ORF expression in different subcellular localizations indicates the different roles of viral ORFs in infected cells. To gain better insights into the function of bait proteins, we performed correlation analysis (Knight *et al*, 2017) of the interactions between the viral bait and host bait proteins, since bait proteins cooperating for a particular function should have similar interaction profiles.

As expected, viral baits with actin filament distribution (Fig EV4A), indicating the cytosolic localization, were not assigned to any specific cluster (Fig 3A). Three highly correlated clusters were detected among the rest of viral baits (Fig 3A). Gene Ontology (GO) enrichment analysis was applied to unique interacting proteins in each cluster (Fig 3B) to highlight the specific functions. The viral baits of cluster 1 were mainly localized in the ER (Fig EV4A), which is contiguous to the nuclear envelope and contains ribosomes. Unsurprisingly, most of the prey proteins of this cluster are involved in RNA processing, nuclear transport, and protein folding (Fig 3C and Dataset EV9). The viral baits of cluster 2 were distributed in several cellular organelles (Fig EV4A), and their abundant interaction partners participated in multiple processes of organization, transportation, and localization of cellular molecules (Fig 3C). Cluster 3 contained the largest group of viral baits, and the unique interaction partners (Fig 3B) in this cluster were significantly enriched in cellular compartment organization and nitrogen compound metabolic processes (Fig 3C). GO enrichment analysis indicated that regulation of host protein localization and cellular compartments is likely a critical strategy for SARS-CoV-2 viral replication.

Similarly, a correlation heatmap (Pearson's correlation) of host bait proteins was generated (Fig 3D). Baits tagged at either the N-terminus or C-terminus appeared in the same cluster (TOP3B and TFRC), except DPP4, for which tagging of the different termini introduced a clear distinction. In the cell, DPP4 exists in two forms: a plasma membrane-bound form and a soluble form (sDPP4, residues 29–766) (Shi *et al*, 2016). The N-terminal affinity tag likely blocks the cleavage of the dipeptide from the N-terminus of DPP4 to generate the membrane-bound form, while the C-terminal tagged DPP4 corresponded to the soluble form (Xi *et al*, 2020). Thus, different cellular contexts led to distinct interaction profiles in different clusters. Bait proteins in the same protein families also clustered together: cathepsins (CTSB, CTSL, and CTSS), the CLEC4 family (CLEC4D and CLEC4E), and the IFITM family (IFITM1 and IFITM3). IFITM family proteins combined with ACE2 and DPP4 to form a larger cluster (Fig 3D and Dataset EV9). This cluster was enriched for specific biological processes: cellular component organization, transport, and localization (Fig 3E). Interestingly, cell surface proteases such as FURIN, TMPRSS2, TMPRSS4, and TMPRSS11, which have been implicated in the cleavage of coronavirus spike proteins (Huggins, 2020; Ou *et al*, 2020), were not grouped into one cluster (Fig 3D), suggesting that their activation occurs through different mechanisms in cells. In summary, analysis of the relationships between baits highlighted the important biological functional groups of bait proteins.

### Viruses target highly connected and central host proteins involved in critical cellular functions

The global analysis of the virus–host interactome suggested that RNA transport, endosomal trafficking, and protein processing in the ER were the most enriched pathways (Dataset EV10). This conclusion is consistent with previous reports. For example, blockage of RNA transport is an effective viral strategy to inhibit the host antiviral response. NSP1 of SARS-CoV or MERS-CoV uses a similar strategy to inhibit host mRNA translation and promote mRNA degradation (Kamitani *et al*, 2009; Lokugamage *et al*, 2015). Our analysis revealed that 42 host proteins interacting with viral baits were involved in RNA transport (Dataset EV10). These proteins are subunits of the TREX complex, exon-junction complex, nuclear pore complex, and translation initiation factors (Fig 4A). Viral interactions with these types of translation machinery control the expression of specific host genes, especially those induced by viral infection. Indeed, 21 proteins out of the 42 interaction partners also appeared in the host–receptor interactome (Dataset EV10).

In the cellular context, an efficient and robust way for viruses to manipulate infected cells is to target critical signaling pathways or well-connected hub proteins in host cells. Therefore, finding these key pathogen–host interaction hubs or pathways is of particular interest. We considered the intersecting macromolecules in the two PPI networks (virus–host and host–receptor) to be likely hub proteins (Fig 4B). 693 proteins were connecting both viral bait and host bait proteins. Furthermore, these proteins had more interconnections with other host proteins (hub protein median: 381 vs. non-hub protein median: 126, *P*-value: $1.71 \times 10^{-91}$; Appendix Fig S10A), suggesting them prominent signaling hubs. Of note, seven of these 693 proteins were among the top 50 proteins with the highest number of reported human protein–protein interactions. Since

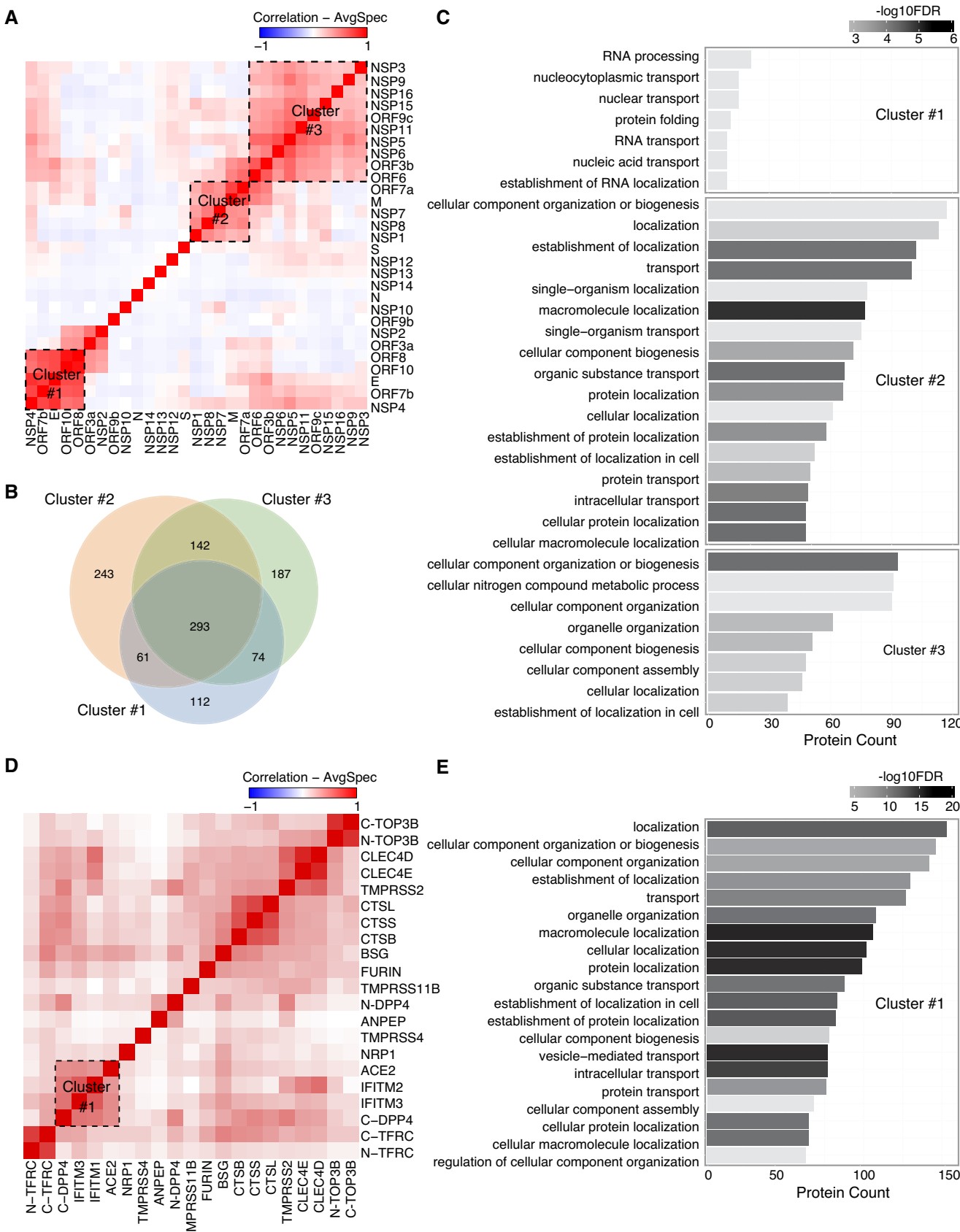

**Figure 3.**

**Figure 3. Bait–bait correlation analysis reveals molecular function clusters.**

The correlation analysis (Pearson correlation coefficient) of bait–bait interaction profiles was performed based on average spectral counts (AvgSpec) (Liu *et al*, 2010) and visualized via a hierarchical clustering heatmap. Direct PPIs connecting bait proteins that form clusters on the heatmap are highlighted for several clusters.

A Viral bait–bait correlation.
B The overlap of proteins in each cluster is shown in a Venn diagram.
C A bar diagram of the enriched pathways associated with the clusters shown and ranked by *P*-value.
D Host–receptor bait–bait correlation; the N- and C-terminal tags on the same bait are indicated.
E The bar chart represents the association of cluster 1 with specific pathways.

protein connectivity is the sole determinant of protein essentiality among all the structural features of the PPI network (He & Zhang, 2006), we designed a simple strategy to score prey protein in SARS-CoV-2 interactome by incorporating our virus–host AP-MS and BioID-MS datasets and four other proteomic datasets (Gordon *et al*, 2020a; preprint: Laurent *et al*, 2020; preprint: Samavarchi-Tehrani *et al*, 2020; preprint: Stukalov *et al*, 2020). The compiled list contained 4,477 viral targeting proteins (Dataset EV11). The frequency was counted based on the number of viral baits which the individual prey has in the SARS-CoV-2 interactome. This frequency was then adjusted by the occurrence of the prey proteins in different SARS-CoV-2 datasets. Interestingly, although the 693 proposed hub proteins covered only about one-sixth of all prey proteins targeted by SARS-CoV-2, most of these proteins were with high adjusted frequency (Appendix Fig S10B), reflecting their functional essentiality by topological importance in the viral processing.

We further identified the overrepresented functional modules of these 693 intersecting host proteins with KEGG pathway analysis (Dataset EV10). In particular, two pathways, namely, endocytosis and protein processing in the ER, are also biologically enriched during virus processing (Dataset EV10). The following section describes those in more detail to illustrate how virus processing affects the host cellular environment.

**Functional characteristics of hub proteins underlying viral processes**

Endosomal entry mechanisms provide many advantages to the virus by allowing SARS-CoV-2 to efficiently spread while avoiding host immunological surveillance (preprint: Bayati *et al*, 2020). We detected the viral ORFs interact with 47 host proteins on the endocytosis pathway. Ten viral ORFs had more than 10 interactions with the proteins of this pathway, including M (27 interactions), Orf3a (26), ORF7a (19), NSP6 (18), S (15), ORF3b (12), E (11), NSP7 (11), ORF10 (11), and NSP10 (10) (Fig 4C).

Viral protein intake (E, M, NSP3, 4, 6, 9, 13, 14, ORF3a, 7a) is regulated by clathrin/AP2 complex-mediated endocytosis with CDC42 (Fig 4C) (Swaine & Dittmar, 2015; preprint: Bayati *et al*, 2020), after which viral particles (NSP5, 6, 7, 16, ORF3a, 7a) end up in RAB5-containing early endosomes (Fig 4C). The very high occurrence of the membrane protein TFRC in both virus–host and host–receptor PPI networks (Fig 4C), together with recent research data (McLaughlin *et al*, 2020; preprint: Tang *et al*, 2020), suggests that TFRC could be additional receptor for SARS-CoV-2 entry. From early endosomes, the virus is transported to late endosomes and eventually to lysosomes (Fig 4C) (Ghosh *et al*, 2020). The internalized virus is uncoated, and the SARS-CoV-2 genome is released from the lysosome/late endosome (Fig 4C) (Bergant Marušič *et al*, 2012; Zhao

*et al*, 2021). The viral RNA replicates and is translated to generate the NSPs and structural components of the virus. Depending on the cell type, RAB7, RAB8, RAB11, and RAB35, which were frequently present in the interaction networks (Fig 4C), were all implicated in the excretion of viral particles (Alenquer & Amorim, 2015).

On the protein processing in the endoplasmic reticulum pathway, the viral proteins interact with 41 of the pathway components, and nine viral ORFs have more than ten interactions with them. The M proteins interact with 27 of the pathway proteins, ORF7a (23), ORF3a (21), ORF10 (21), S (20) ORF8 (17), NSP6 (15), E (12), and ORF6 (11) (Fig 4D).

The coronavirus infection disturbs ER homeostasis and causes ER stress (Fung & Liu, 2014), which plays a vital role in innate immune signaling in response to infections (Fung *et al*, 2014; Shi *et al*, 2019). The structural proteins of coronavirus (S, E, and M) go through post-translational modifications in the ER, triggering ER stress and inducing the expression of ER chaperones such as CANX, CALX, HSPA5, PDIA3, and PDIA4 (Fig 4D) (Ma *et al*, 2008; Siu *et al*, 2008; Fukushi *et al*, 2012). Under acute or prolonged ER stress, cell apoptosis is activated. Moreover, the formation of coronavirus (NSP3, NSP4, and NSP6)-induced double-membrane vesicles, a hallmark of coronavirus replication, is promoted by COPII-coated secretory vesicles or proteins (SEC23/24 proteins, LMAN1, SAR1A) (Fig 4D) (Wolff, Limpens, *et al*, 2020; Wolff, Melia, *et al*, 2020). On the other hand, overexpression of viral ORFs in host cells can lead to the accumulation of misfolded proteins. Therefore, the ER-associated degradation (ERAD) pathway (e.g., NGLY1, HSPA5, STUB1, BAG2, HAP90B1, RNF5) (Fig 4D) is activated.

Although the precise mechanism remains elusive for SARS-CoV-2, overlaps between the viral interactome and receptor interactome indicate a core of human proteins involved in cellular response to viral entry, and infection and the viral capacity to hijack the cell machinery for viral replication. These highly targeted hub proteins with significant biological functions were prioritized for further *in silico*-druggability *investigation*.

**Discovery of potential antiviral drug targets and candidate drugs**

The vast complexity of the preclinical drug design and clinical trials makes drug development a time-consuming process, prolonging this pandemic. To speed up the process, our effort to find promising antiviral drug candidates is focused on drug repositioning. A useful *in silico* tool for finding candidates for repurposing is structure-based virtual screening.

To identify drug targets among the 693 hub proteins for virtual screening (Fig 4B), we further analyzed these proteins based on the following two criteria: (i) availability of a ligand-protein crystal structure with a relevant binding pocket, and (ii) the pocket

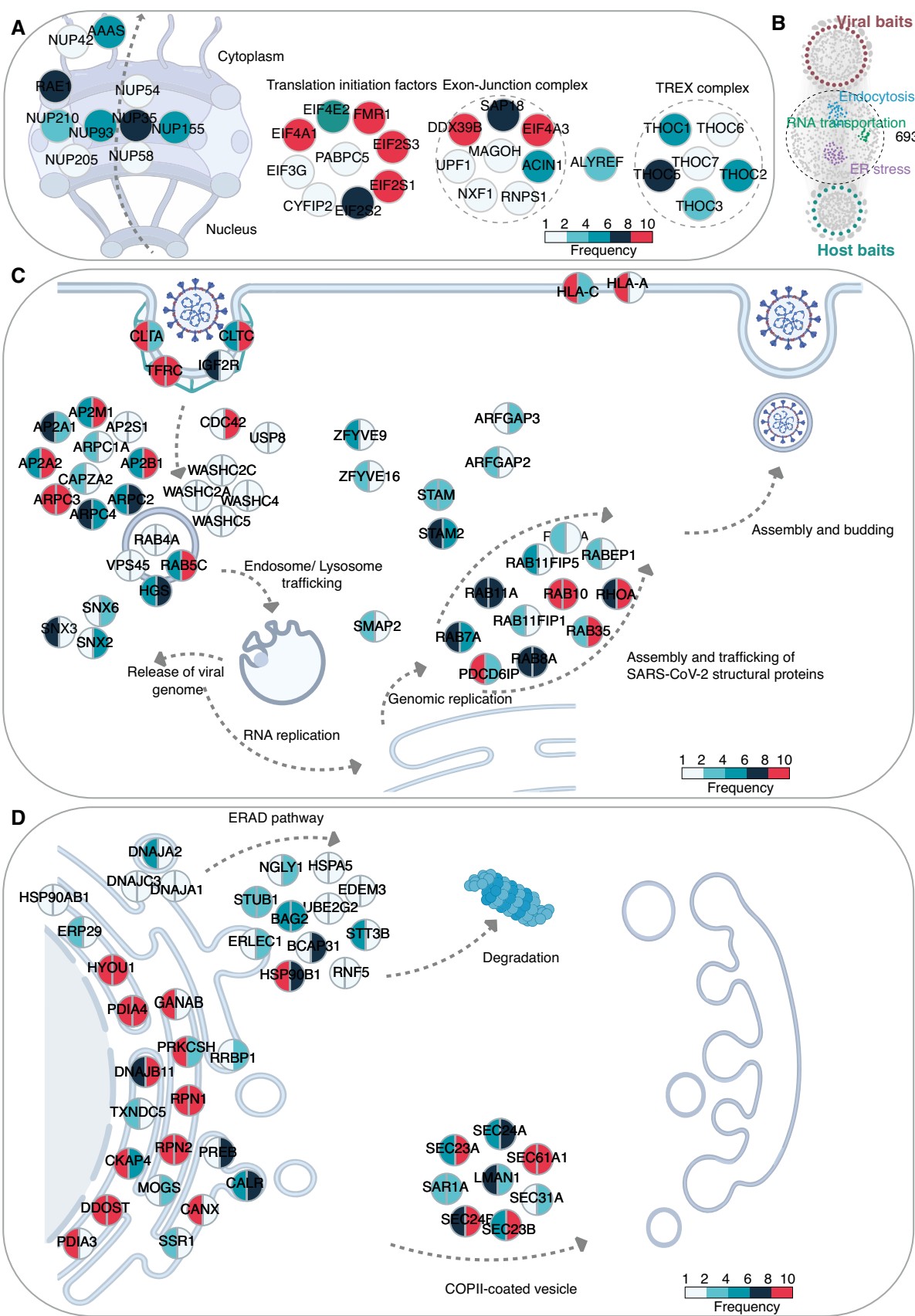

**Figure 4.**

◄

**Figure 4. Enrichment analysis for specific cellular pathways.**

A   KEGG pathway enrichment analysis of the virus–host PPI network was conducted, and RNA transport-related proteins were highly enriched. The color of a node corresponds to the frequency of occurrence of the prey that has been detected in the virus–host interactome.

B   In total, 693 proteins were connected by both viral baits and host–receptor baits in the combined interactome. KEGG pathway enrichment analysis of these 693 proteins was conducted, and three pathways, namely, RNA transportation, endocytosis, and protein processing in the ER, were highlighted.

C, D   Enriched proteins were illustrated by the schematic diagram relevant to endocytosis (*P*-value: $5.84 \times 10^{-15}$) (C) and protein processing in ER (*P*-value: $1.80 \times 10^{-22}$) (D). The color of each slice in a node corresponds to the frequency of occurrence of the prey that was detected in virus–host interactome (left half slice) and host–receptor interactome (right half slice).

druggability of the selected protein structure (Fig 5A). Ligand binding to such pockets can induce conformational changes that affect interactions with other protein interaction partners of the target (Dauch *et al*, 2016; Rudalska *et al*, 2021). Thus, this approach has the potential to disrupt interactions between hubs and viral proteins.

Ligand selection started with the semiautomated curation of four well-established databases: DrugBank (Knox *et al*, 2011), ChEMBL (Bento *et al*, 2014), NPC browser (Huang, Southall, *et al*, 2011), and SciFinder database (scifinder.cas.org). After filtering and *in silico* preparation, a library of 5,518 unique, approved, investigational, and withdrawn drugs were considered as candidates for repurposing in our virtual screening (Fig 5A).

We performed molecular docking simulations of the compound library against selected proteins to model and evaluate possible binding poses for each compound. The binding affinity between the target and the drugs was first estimated with the Glide standard precision (SP) docking protocol, and the top-scoring 20% of the hits were then subjected to extra precision (XP) docking to select 200 top-scoring compounds. The final selection of up to eight virtual hits per target was based on the interactions between the receptor and top-ranked ligands and favorable ligand geometry. Overall, the process suggested 59 compounds that could be repurposed for 15 protein targets (Fig 5B and Dataset EV12). Most of these candidate drugs had antiviral, anti-neoplastic/cancer, anti-inflammatory, or antibacterial activities (Fig 5B and Dataset EV12). Among those, three antidiabetic agents (licogliflozin, pioglitazone, and MK-0767) were covered and may have a potential effect on SARS-CoV-2 infection based on our screening results. Pioglitazone treatment was for instance found beneficial to a diabetic patient with COVID-19 (Jagat *et al*, 2020), and the compound is under further clinical evaluation for the treatment of diabetes type 2 during COVID-19 infections.

Moreover, we identified highly connected subnetworks with biological relevance for viral processing (Fig 4B). Repurposing drugs to target proteins from these core subnetworks may be explored to reduce disease severity more effectively. Some virtual hits were shared by more than one target, such as guadecitabine for mitochondrial Rho GTPase 2 (RHOT2) and the GTP-binding protein SAR1a (SAR1A) or orilotimod for nucleoside diphosphate kinase 3 (NME) and leucyl cystinyl aminopeptidase (LNPEP) (Fig 5 B). This indicates their similar ligand-binding properties which might be exploited to simultaneously target more than one of the important hub proteins by a single drug. The repurposing of several other candidate drugs is supported by previous studies. For example, SNX-5422, targeting endoplasmin (HSP90B1), which is in charge of protein processing in the ER (Figs 4C and 5C), was suggested for use as an oral post-exposure prophylactic or early-

phase therapeutic for SARS-CoV-2 infection (preprint: Goswami *et al*, 2021). The Ras-related proteins RAB1 and RAB7 are involved in the endocytic pathway (Fig 4C), and their activation is modulated to facilitate SARS-CoV-2 entry and intracellular survival (Sicari *et al*, 2020; Daniloski *et al*, 2021). Their potential inhibitors, ritobegron (Maruyama *et al*, 2012) and pemetrexed (Wandinger-Ness *et al*, 2014), may therefore prevent the entry of the virus (Fig 5C). These compounds are of high interest for further studies on how virus infects the different cell types during the infection; however, their potential cytotoxicity and side effects should be carefully considered when selecting the potential drugs for clinical investigations to treat the COVID-19. Taken together, our results constitute a list of compounds that can potentially be effective against SARS-CoV-2.

### SARS-CoV-2 modulates rod and ring formation or assembly

Ribavirin was identified as one of the top hits for the hub protein hexokinase 1 by virtual screening (Fig 5B). Although ribavirin shows antiviral activity against a variety of RNA viruses, including influenza virus (Eriksson *et al*, 1977), poliovirus (Crotty *et al*, 2000), and hepatitis C virus (HCV) (Pawlotsky *et al*, 2004), it remains debatable whether ribavirin is an effective treatment for COVID-19 (Casaos *et al*, 2019; Hung *et al*, 2020; Khalili *et al*, 2020; Tong *et al*, 2020). The molecular mechanism underlying the action of ribavirin has remained unclear. One of the interesting findings from *in vitro* cell-based models is that ribavirin treatment can strongly induce the rapid formation of helical polymers composed of stacked IMP dehydrogenase (IMPDH) octamers (Fig EV5A), referred to as "rod and ring (RR)" structures (Thomas *et al*, 2012; Keppeke *et al*, 2019) (Fig EV5B). Anti-RR antibodies have been detected in HCV patients only after ribavirin treatment, not prior to treatment (Alsius *et al*, 2019). This implies that the RR structure could be used as an indicator of ribavirin efficacy.

The stable cell line inducibly expressing IMPDH2 formed RRs upon ribavirin addition (Fig EV5B). However, the expression of a viral ORF (NSP12) drastically decreased the formation of these structures in response to ribavirin (Fig EV5B). Our results indicated that several viral ORFs (NSP2, NSP4, NSP9, NSP12, and NSP14) with filament distribution (Fig EV4A) interacted with IMPDH (Dataset EV4). Therefore, we hypothesize that the viral ORFs may prevent the ribavirin-induced formation of RRs in host cells by directly or indirectly competing with the associated proteins of IMPDH that promote octamer formation. Our study presents a possible molecular-level mechanism explaining why monotherapy with ribavirin might not improve the SARS-CoV-2-induced mortality rate, as reported in a retrospective cohort study (Tong *et al*, 2020).

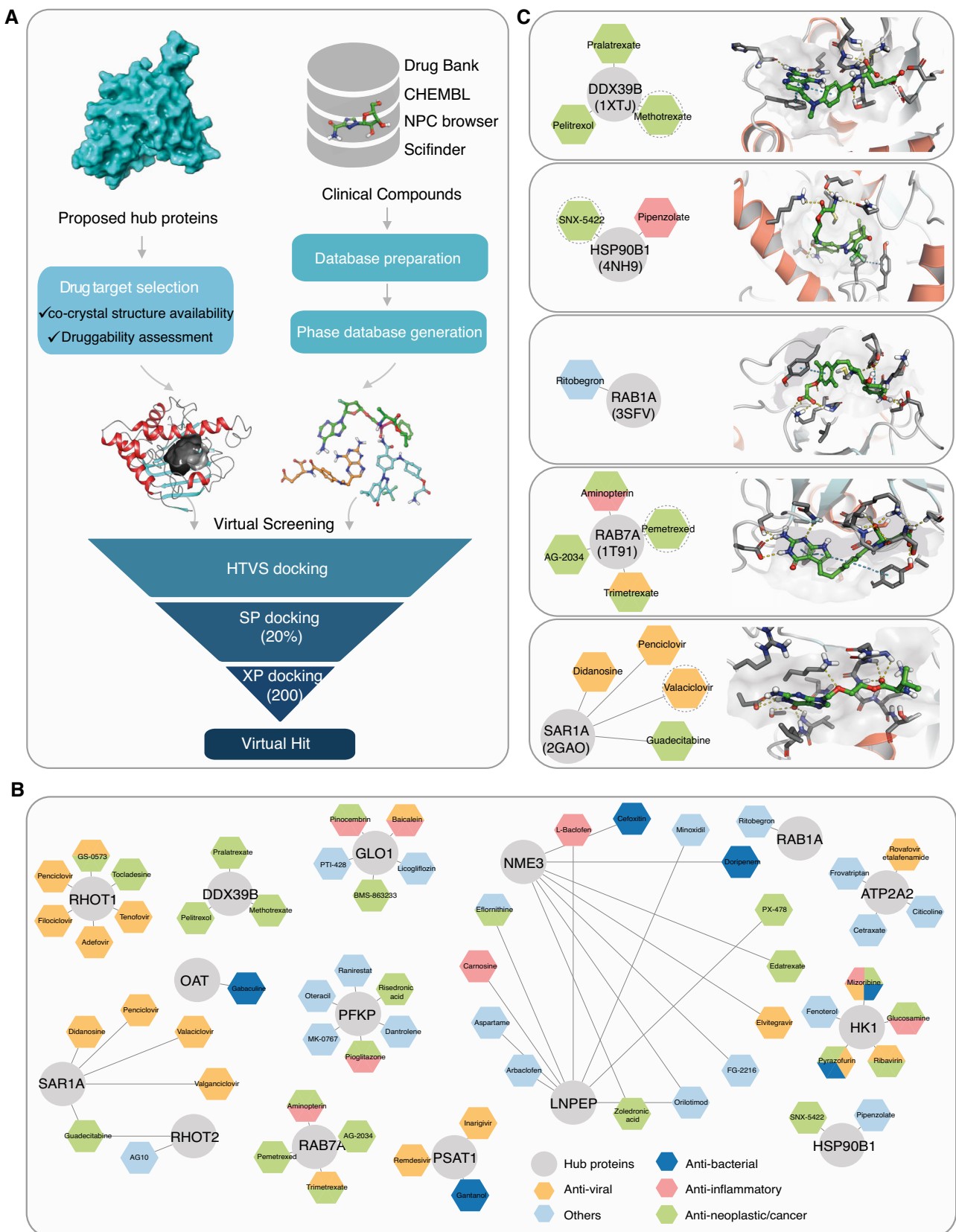

**Figure 5.**

◀

**Figure 5.   Interaction network-based drug candidates for repurposing.**

A   Diagram illustrating the workflow of PPI network-based drug repurposing studies, where we used proposed hub proteins for the *in silico-druggability* investigation.
B   The simplified compound-target network with only the primary proteins (circles) and the potential drug candidates (hexagon) are shown and colored according to drug activities.
C   Putative bound orientations of ligands were predicted for the protein model with the given PDB ID and selected drug candidates.

## Antiviral activities of proposed repurposing candidates

We next investigated the antiviral activity of the proposed repurposing candidates, using an image-based drug screen with infectious SARS-CoV-2, detected by immunostaining of N protein. Ten drugs available in house were evaluated in this study (Dataset EV13). Among them, six drugs (baicalein, methotrexate, guadecitabine, PX-478, mizoribine, and BMS-863233) showed some potential activities against SARS-CoV-2 (Fig 6A–C and Appendix Fig S11), as defined by the area under the curve (AUC) and half-maximal inhibitory concentration (IC50) values (Dataset EV13). *In vitro* antiviral effects have been previously described for the majority of those compounds (Dataset EV12). Interestingly, to our knowledge for the first time, experimental results suggested the potential application of guadecitabine against SARS-CoV-2. This should be assessed further for potential clinical use.

In line with our results, methotrexate (MTX) has shown antiviral effects on SARS-CoV-2 infected Vero E6 cells *in vitro* (Caruso *et al*, 2021). This raises the question of whether its antiviral effect can be related to the inhibition of PPIs of its putative drug target, DEAD-box (DDX) RNA helicase DDX39B. Proteomics and Co-IP screens showed DDX39B can interact with both viral ORFs (NSP5, 6, 13, 14) and human receptors (TMPRSS2, TOP3B, CTSL) (Fig EV3 and Dataset EV6). Therefore, we performed a cell-based co-immunoprecipitation assay (Fig 6D) to which MTX was added.

DDX RNA helicases are required for replication of a number of human viral pathogens, including HIV-1 (Fang *et al*, 2004), influenza A (Diot *et al*, 2016), infectious bronchitis virus (IBV)-CoV (Xu *et al*, 2010), and SARS-CoV (Chen *et al*, 2009). These viruses hijack host helicases, to support reverse transcription, and nuclear exportation (Squeglia *et al*, 2020). DDX39B has been linked to the inflammatory response (Szymura *et al*, 2020), but little is known about its potential role in SARS-CoV-2 replication. Quantitation of Co-IP showed a trend toward decreased binding of DDX39B with viral ORFs NSP13, NSP14, S, and host receptors TMPRSS2 in the presence of MTX (Fig 6E). NSP13 and NSP14 are the key components of the replication-transcription complex (RTC) of SARS-CoV-2. Therefore, MTX could inhibit virus replication. On the other hand, MTX inhibits interactions of DDX39B with S protein and TMPRSS2 (Fig 6E), indicating that it may act against viral entry during the early stage of infection.

However, not all the antiviral effective drugs were able to show the effects on protein interaction level, such as BMS-863233, for which we did not observe any effects on PPI level (Appendix Fig S12). This could be because many drugs have a wide range of target molecules or are otherwise toxic to the host cells in higher concentrations (Fig 6C). Thus, more specific targeted inhibitors should be used for further investigation of virus–host interactome.

In summary, in the current study, we present a plethora of host–pathogen protein–protein interactions involved during the SARS-CoV-2 infection. Inhibition of these interactions can pave the way for promising novel approaches to rational drug design focused on disrupting these key PPIs.

## Discussion

In this work, we conducted a large-scale proteomic study that identified 4,781 unique high-confidence virus–host PPIs using 29 viral ORFs and 4,362 unique HCIs for 18 suggested receptors/proteases/cofactors for SARS-CoV-2. Subsequent subcellular localization and bait correlation analysis revealed the bait similarity profiles of specific functional clusters. Furthermore, by integrating the virus–host interaction network and host–receptor interaction network, a comprehensive evaluation pointed to several host factors needed for viral entry, replication, or spread. Finally, we suggested candidate drugs targeting specific proteins for treatment via a drug repurposing approach. In addition, using a cell-based assay, we demonstrated for one of the candidates, MTX, that the antiviral effect can be related to disrupting the specific protein interactions of the targeted hub protein DDX39B.

SARS-CoV-2 utilizes diverse cytoplasmic structures for processing, with two possible consequences: triggering the antiviral response to protect host cells or allowing the virus to escape the cellular response, and thereby facilitating virus replication. NSP3 can induce nuclear actin polymerization to assemble "ring" structures in the host cell. The perinuclear structure may function as a shield to protect host cell genome integrity during virus-induced instability within the cell. On the other hand, the N protein prevents the formation of stress granules in the host cell and enables the synthesis of viral proteins using the host translation machinery (McCormick & Khaperskyy, 2017; preprint: Samavarchi-Tehrani *et al*, 2020). In this study, we found that viral ORF expression could prevent RR formation. This observation suggests that these structures can be subverted to enhance viral replication. Overall, the recognition of viral ORFs and host cellular proteins relevant to such cellular structures will significantly advance the development of cell-based assays for antiviral drug screening.

We first focused on the virus–host interactome. Functional annotation and network-based analyses of virus–host PPIs highlighted cellular pathways and biological functions targeted by the virus, including RNA transport, endocytosis, and protein processing in the ER. Furthermore, any perturbation in host homeostasis caused by viruses is dispersed throughout the host PPI network. Thus, it is necessary to study the virus–host interactome in the context of the host PPI network. However, for nine selected bait proteins used in this study, less than 50 known interactions were retrieved from the latest and most comprehensive human PPI databases. Therefore, our study fills the gap and provides the first specific human interactome dataset highly relevant to SARS-CoV-2. The integrated network

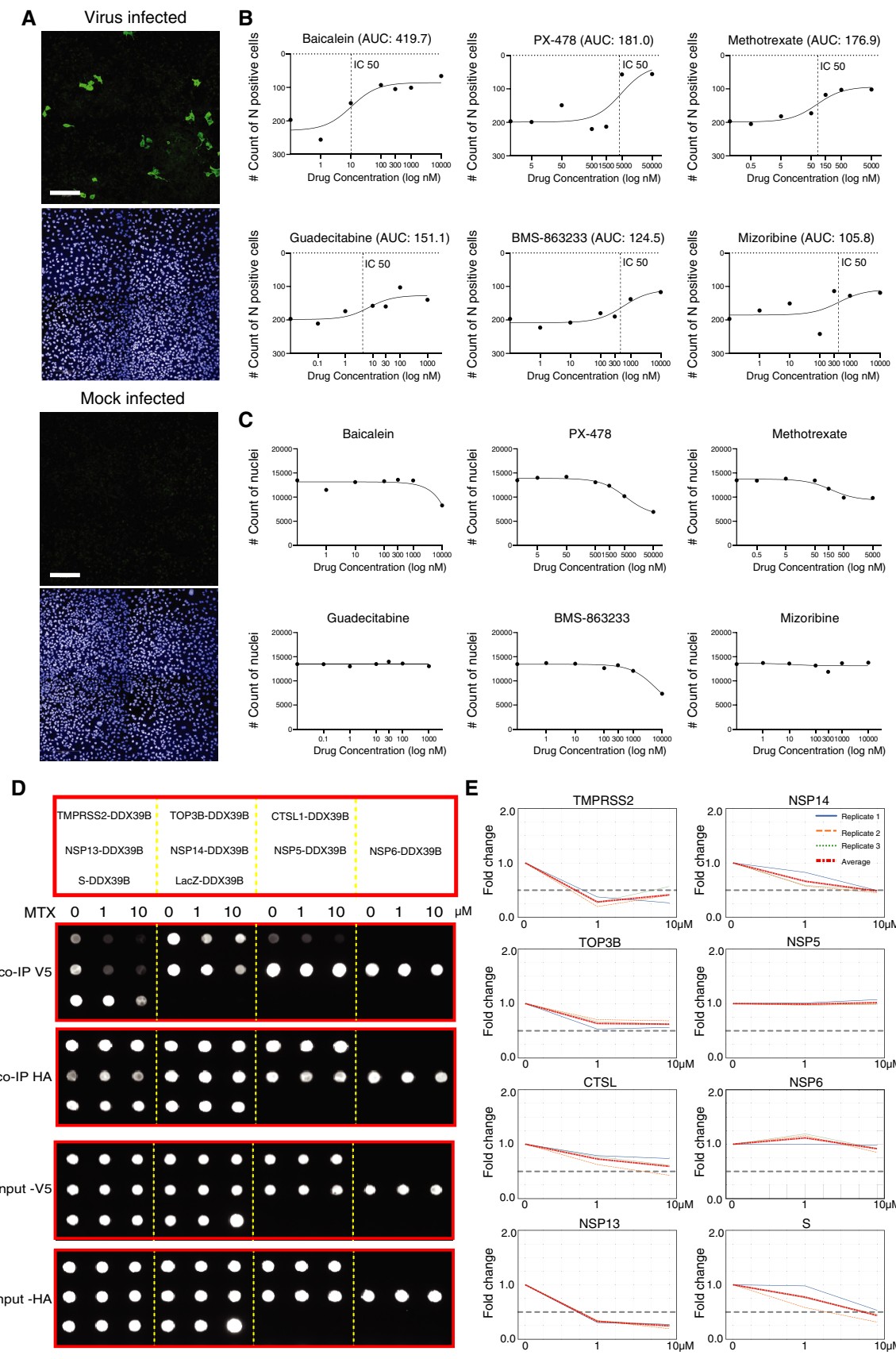

**Figure 6.**

**Figure 6.   The antiviral activities of selected drugs against SARS-CoV-2 *in vitro*.**

A   For the image-based drug screening, Calu-1 cells were infected with SARS-CoV-2 in the presence of drugs with indicated doses for 48 h, immunostained, imaged, and analyzed. The example images of mock-infected cells and virus-infected control cells (no drugs). The viral yield in cells was detected with an antibody for the viral N protein (virus-infected cells; upper image), and nuclear staining (mock-infected cells; lower image) (Scale bar: 200 μm).

B   The dose–response curves for six drugs with potential antiviral effects. The drug-induced inhibition of virus infection (count of N protein-positive cells) was quantified using the area under the curve (AUC) values.

C   The effect of drugs on cell viability in the presence of the virus, e.g., cytotoxicity, was evaluated by the count of nuclei.

D   Inhibition of interactions formed with DDX39B in the presence of MTX using the Co-IP assay. DDX39B is fused to the C-terminus of a V5 tag, and the interaction partner is fused with a Strep-HA tag. The diagram (upper) shows the interaction pairs formed with DDX39B used for Co-IP assay. Dot-blot results (lower) show the interaction pairs (arrangement as upper panel) in the presence of different concentrations of MTX as indicated on the top of the image. The image displayed is representative of three replicates ($n = 3$).

E   Quantification of DDX39B binding to different interaction partners. Quantification of V5-tagged DDX39B protein amount was normalized over HA-tagged interaction partner intensity and displayed as a fold change over the ratio of untreated cells. Data that represent the mean of three replicates are highlighted in red. A fold change > 0.5 is considered significant.

analysis described herein has expanded our understanding of human–virus interactions from a unique perspective, which is different from previous analyses based either only on the virus–host interactome or the virus–host interactome integrated with the entire human PPI network. The analysis also enabled us to pinpoint multiple central hub proteins that appear to be particularly relevant to the critical pathways for viral replication.

We acknowledge several limitations of this current study. Firstly, similarly as in the four mentioned large-scale studies, we analyzed the individual interactomes in uninfected cells. The pathophysiological and molecular perspective in the context of SARS-CoV-2 infection requires further functional validation. Secondly, we focused on highly connected hub proteins for drug target modeling. However, hub proteins are not always functionally essential and are involved in multiple biological processes. Caution is also required, since inhibition or activation may lead to adverse side effects for the host. Furthermore, ligand binding to orthosteric pockets may not result in the desired conformational changes to disrupt the interaction with viral proteins. Thirdly, we highlighted ER-calnexin pathway in our study to address the possible relationship with viral processes.

However, overexpression of MAC-tagged bait protein could also induce the unfolded protein response, which results in the up-regulation of genes encoding ER-resident chaperones such as calnexin, calreticulin, and hypoxia-upregulated 1 gene (HYOU1) (Lindholm *et al*, 2017). Although these proteins are not often or systematically detected in large-scale interactome studies, further experiments will be needed to rule out the possibility. Fourthly, selected repurposing candidates were validated using an *in vitro* cell-based assay. The efficacy evaluation of these drugs against SARS-CoV-2 will require extensive follow-up work in other relevant cell lines or animal models. Finally, our drug repurposing candidates rely largely on monotherapy, that is, the use of a single drug against one protein target. Combination therapy, which uses multiple drugs for several targets, should be further considered to improve clinical outcome. Our protein interaction data provide an excellent resource for network-based drug combination design strategies.

In summary, our study provides a comprehensive and complimentary overview of the SARS-CoV-2 host protein interactome. The findings highlight 693 hub proteins as potential therapeutic targets and pave the way to the identification of effective antiviral drugs.

# Materials and Methods

### Reagents and Tools table

| Reagent/Resource | Reference or Source | Identifier or Catalog Number |
| --- | --- | --- |
| **Experimental Models** | | |
| HEK Flp-In™ 293 T-Rex cell | Life Technologies | B6.129P2Gpr37tm1Dgen/J |
| HEK293 | ATCC® | CRL-1573™ |
| Calu-1 | ATCC® | HTB-54 |
| U2-OS | ATCC® | HTB-96™ |
| **Recombinant DNA** | | |
| pOG44 Flp-Recombinase Expression Vector | Life Technologies | V600520 |
| Gateway™ pDONR221™ | Thermo Fisher Scientific | 12536017 |
| MAC-tag-N destination vector | Addgene | 108078 |
| MAC-tag-C destination vector | Addgene | 108077 |
| MAC-GFP | Addgene | 139636 |

**Reagents and Tools table**   (continued)

| Reagent/Resource | Reference or Source | Identifier or Catalog Number |
|---|---|---|
| pDEST40 | Thermo Fisher Scientific | 12274015 |
| pcDNA5/FRT/TO | Invitrogen™ | V652020 |
| **Antibodies** | | |
| Anti-HA Tag Antibody | Biolegend | PRB-101C |
| Goat anti-Mouse IgG (H + L) Highly Cross-Adsorbed Secondary Antibody, Alexa Fluor Plus 488 | Thermo Fisher Scientific | A32723 |
| Alexa Fluor594-conjugated secondary antibody | Thermo Fisher | A32740 |
| Alexa Fluor-488-conjugated secondary antibody | Thermo Fisher | A-11001 |
| Anti-alpha Tubulin antibody | Abcam | ab7291 |
| Goat anti-mouse IgG H&L (HRP) | Abcam | 97023 |
| anti-V5 antibody | Invitrogen | 37-7500 |
| Alexa Fluor 594-conjugated phalloidin | Thermo Fisher | A32740 |
| Rabbit anti-SARS-CoV2 N protein | Rusanen et al (2021) | |
| Alexa Fluor goat anti-rabbit IgG 488 | Invitrogen | A11008 |
| Alexa Fluor goat anti-mouse IgG 568 | Invitrogen | A11004 |
| α-tubulin monoclonal antibody | Abcam | ab184613 |
| **Chemicals, enzymes and other reagents** | | |
| Gateway™ LR Clonase™ Enzyme Mix | Life Technologies | 11791043 |
| DMEM | Life Technologies | 41965062 |
| HEPES | Sigma | 7365-45-9 |
| Hoechst | Lifetech | 33342 |
| IGEPAL (electrophoresis reagent) CA630 | Sigma | 9002-93-1 |
| Amersham™ ECL™ Prime | Cytiva | RPN2232 |
| FuGENE® 6 Transfection Reagent | Promega | E2691 |
| Pierce™ BCA Protein assay | Thermo Fisher Scientific | 23225 |
| Penicillin–streptomycin | Life Technologies | 15140130 |
| Formic Acid, ≥95% | Sigma | 64-18-6 |
| Tris(2-carboxyethyl)phosphine (TCEP) | Sigma-Aldrich | 51805-45-9 |
| Ammonium bicarbonate (AMBIC) | Sigma-Aldrich | 1066-33-7 |
| N,N-Dimethylformamide | Sigma | 68-12-2 |
| Sodium chloride | Merck | 7647-14-5 |
| Iodoacetamide (IAA) | Sigma-Aldrich | 64-69-7 |
| Ethylenediamine, ≥ 99.5% | Sigma | 107-15-3 |
| Hygromycin B | Invitrogen | 10687010 |
| Laemmli sample buffer | Bio-Rad | 1610737 |
| Sodium dodecyl sulfate | Sigma | 151-21-3 |
| Triton X-100 | Sigma | X100-500 |
| Phenylmethanesulfonylfluoride (PMSF) > 98.5% | Sigma | 329-98-6 |
| water MS-grade | Merck | 7732-18-5 |
| Trypsin-EDTA | Gibco | 25200-56 |
| Sequencing Grade Trypsin | Promega | V5113 |
| Strep-Tactin® Sepharose® 50% (vol/vol) suspension | IBA life sciences | 2-1201-010 |
| Bio-Spin® Chromatography Columns | Bio-Rad | 732-6008 |
| Sodium fluoride | Sigma | 7681-49-4 |
| Protease inhibitor cocktail | Sigma-Aldrich | P8340/P2745 |

**Reagents and Tools table**  (continued)

| Reagent/Resource | Reference or Source | Identifier or Catalog Number |
|---|---|---|
| Biotin | Thermo Fisher Scientific | 29129 |
| Benzonase® Nuclease | Santa Cruz Biotechnology | sc-202391 |
| Tetracycline hydrochloride | Sigma-Aldrich | T3383-25G |
| ECL Western Blotting Detection Reagent | GE Healthcare | RPN2209 |
| DRAQ5 | Thermo Fisher | 62251 |
| DAPI, 4',6-diamidino-2-phenylindole, dihydrochloride | Sigma | D9542 |
| DAPI, 4',6-diamidino-2-phenylindole, dihydrochloride | Santa Cruz | sc-3598 |
| RPMI medium | Sigma | R0883-1L |
| Methotrexate | Selleck Chemicals | CL-14377, CAS: 59-05-2 |
| Compound XL413 | Selleck Chemicals | BMS-863233,1169562-71-3 |
| Restore Plus Stripping buffer | Thermo Fisher | 46430 |
| Fetal bovine serum (FBS) | Gibco | 10270-106 |
| Skimmed milk powder | Valio | D1-5824 |
| Ribavirin | Sigma-Aldrich | 36791-04-5 |
| TWEEN®20 | Sigma-Aldrich | P1379-250ML |
| Pure nitrocellulose membrane 0.45 μm | Perkin-Elmer | NBA085C001EA |
| SDS-PAGE gel | Bio-Rad | 4561096 |
| Bradford reagent | Bio-Rad | 500-0006 |
| **Software** | | |
| Cytoscape v3.4.0 | http://www.cytoscape.org | Shannon *et al* (2003) |
| nSolver™ 4.0 | Nanostring Technologies | |
| SlideBook | 3I Intelligent Imaging Innovations | |
| ImageJ | https://imagej.net/software/fiji/ | |
| SCT BIAS software | Single-Cell-Technologies | Daly *et al* (2020) |
| Xcalibur 2.0.7 SP1 | Thermo Fisher | |
| MaxQuant version 1.6.0.16 | https://www.maxquant.org/ | Cox and Mann (2008) |
| Significance Analysis of INTeractome 3.6.0 | http://saint-apms.sourceforge.net/Main.html | Choi *et al* (2011), Choi *et al* (2012) |
| Proteome Discoverer | Thermo Fisher | |
| Prism 9.2 | GraphPad Software | |
| Harmony 4.9 | PerkinElmer | |
| **Other** | | |
| Easy-nLC II Liquid Chromatography system | Thermo Fisher Scientific | LC120 |
| Q Exactive™ Hybrid Quadrupole-Orbitrap™ Mass Spectrometer | Thermo Fisher Scientific | IQLAAEGAAPFALGMAZR |
| Electrospray ionization sprayer | Thermo Fisher | |
| Fluorescence microscope | Leica | Leica TCS SP8 STED |
| Turbo transfer system | Bio-Rad | |
| Nanostring nCounter® | NanoString Technologies, Seattle, USA | |
| NanoString nCounter gene expression platform | NanoString Technologies, Seattle, USA | |
| nCounter Prep Station | NanoString Technologies, Seattle, USA | |
| Intelligent Imaging Innovations (3i) Marianas inverted spinning disk microscope | Yokogawa | Yokogawa CSU-X1 M1 5,000 rpm |
| Leica Stellaris 8 Falcon microscope | Leica | |

**Reagents and Tools table**  (continued)

| Reagent/Resource | Reference or Source | Identifier or Catalog Number |
|---|---|---|
| Opera Phenix spinning-disk confocal microscope | PerkinElmer | |
| C-18 Macro Spin Colum | Nest Group | 800-347-6378 |
| Bio-Dot® Microfiltration System | Bio-Rad | 1703938 |
| iBright Imaging Systems | Thermo Fisher | |
| Acoustic liquid handler Echo 550 | Labcyte | |
| RNeasy® Mini Kit | Qiagen, Hilden, Germany | 74004 |

## Methods and Protocols

### Cloning

Entry clones containing genes of interest were obtained from the ORFeome collection (ORFeome and MGC Libraries; Genome Biology Unit supported by HiLIFE and the Faculty of Medicine, University of Helsinki, and Biocenter Finland). The gene of interest used for generating the stable cell lines was either fused to a MAC-tag-N (Addgene, Plasmid #108078) or MAC-tag-C (Addgene, Plasmid #108077) destination vector using Gateway cloning techniques. All the plasmids used in this project are available from Addgene (https://www.addgene.org/Markku_Varjosalo/). The gene of interest used for protein interaction validation was fused to either a modified pDEST40 vector (with a 3×V5 C-terminal tag) (Varjosalo *et al*, 2008) or a modified pcDNA5/FRT/TO destination vector (with a Streptavidin-binding peptide and HA tag on either C-terminus or N-terminus) (Glatter *et al*, 2009) using Gateway cloning techniques.

### Cell culture

The Flp-In™ 293 T-REx cell line (Invitrogen, R78007), HEK293 cell line (ATCC® CRL-1573™), and U2-OS (ATCC®, HTB-96™) were routinely maintained under the manufacturer-recommended conditions. Transfection and generation of stable cell lines were performed as previously described (Liu *et al*, 2020). Briefly, Flp-In™ 293 T-REx cells were co-transfected with the MAC-tagged bait or GFP construct and pOG44 vector (V600520, Invitrogen) using FuGENE 6 transfection reagent (Promega, Wisconsin, USA). Forty-eight hours post-transfection, cells were selected in medium containing hygromycin B (100 μg/ml; Invitrogen) for 3 weeks. The positive clones, containing stable isogenic MAC-tagged baits, were further expanded to 80% confluence in 30 × 150-mm cell culture plates (CELLSTAR, Greiner) in complete culture medium. Cells from five plates were used for one biological replicate, in which 1 μg/ml tetracycline (Sigma-Aldrich, T3383-25G) with/without 50 μM biotin (Thermo Fisher, 29129) was added for 24 h before harvesting. Three biological replicates were generated for each condition. The Calu-1 cell line (ATCC® HTB-54) used for drug screening was routinely maintained in RPMI medium (Sigma, R0883-1L), containing 10% FBS (Gibco, 10270-106), 1% L-glutamine, and 1% solution of penicillin–streptomycin. Subculturing was performed twice a week using 0.25% trypsin-EDTA (Gibco, 25200-56).

### Western blot analysis

Cell lysate samples were obtained before loading on beads for protein purification. Protein concentrations were determined with Bradford reagent (Bio-Rad, 500-0006) and adjusted to equal concentration with Laemmli Sample Buffer (Bio-Rad, California, USA, 1610737). Samples were loaded and run on 4–20% gradient SDS–PAGE (Bio-Rad, 4561096). Proteins were transferred to nitrocellulose membrane with the Trans-blot Turbo transfer system (Bio-Rad). The membrane was blocked in 5% milk-TBS with 0.1% Tween-20 (TBS-T) for 1 h at RT. Primary antibodies were diluted into fresh blocking buffer overnight at 4°C. The following primary antibodies were used as follows: HA mouse monoclonal antibody (BioLegend, PRB-101C, 1:2,000 dilution) and anti α-tubulin monoclonal antibody (Abcam, ab184613, 1:1,000 dilution). The membrane was probed the following day with HRP-linked secondary antibodies (1:2,000) using ECL reagent (GE Healthcare, RPN2209).

### NanoString platform for immune response profile

The Flp-In™ 293 T-REx cells and generated viral ORF stable cells were harvested and used to obtain the necessary amount of RNA (~100 ng) using a RNeasy® Mini Kit (Qiagen, Hilden, Germany, 74004). The RNA was added to the NanoString nCounter gene expression platform (NanoString Technologies, Seattle, USA) and hybridized using the NanoString nCounter® Human v1.1 PanCancer Immune Profiling Panel (770 transcripts) according to the manufacturer's protocol. Hybridization processing was performed on the nCounter Prep Station (NanoString Technologies), and the pre-pipetted cartridges were scanned in triplicate on the NanoString Digital Analyzer. Gene expression data were analyzed using nSolver™ 4.0 analysis software (NanoString Technologies). Positive controls were used to adjust for possible variations in samples, and subsequent data normalization was performed by using housekeeping genes present in the panel and by applying negative control subtraction. Log2 transformation was applied to normalize gene counts to calculate the fold changes between the samples and untransfected control cells.

### Immunofluorescence

U2-OS cells were transfected with the specified viral bait ORF and fixed in 4% paraformaldehyde in PBS. Bait proteins were detected with the anti-V5 antibody (Invitrogen, 37-7500, 1:1,000 dilution), followed by Alexa Fluor 488-conjugated secondary antibody. Actin was stained with Alexa Fluor 594-conjugated phalloidin (Thermo Fisher), and nuclei were stained with DAPI (Santa Cruz) or DRAQ5 (Thermo Fisher). After staining, the cells were washed twice and imaged in PBS. Imaging was performed using an Intelligent Imaging Innovations (3i) Marianas Inverted Spinning Disk (Yokogawa CSU-X1 M1 5,000 rpm) microscope, equipped with Andor Neo sCMOS camera and solid-state lasers (405 nm/100 mW, 488 nm/150 mW,

561 nm/50 mW, 640 nm/100 mW). Cells were imaged using a 20 ×/0.8 Plan-Apochromat Ph2 WD = 0.55 M27 objective lens (Zeiss). NSP3-expressing cells were further imaged with Leica Stellaris 8 Falcon microscope, using HyD X and HyD S detectors, diode 405 and WLL SuperK Extreme lasers, and a Leica HC PL APO CS2 63×/ 1.20 W objective lens. The voxel size was 0.068 × 0.068 ×0.356 μm; Pinhole 1 AU.

HEK 293 cell stable stably expressing MAC-tagged IMPDH2 was transfected with the V5-tagged NSP12 or GFP vector and treated with 1 μg/ml tetracycline (Sigma-Aldrich, T3383-25G) and 2 mM ribavirin (Sigma-Aldrich, 36791-04-5) for 24 h before fixation. IMPDH proteins were detected using an anti-HA antibody (Thermo Fisher, 26183, dilution 1:1,000 dilution) followed by an Alexa Fluor 594-conjugated secondary antibody (Thermo Fisher, A32740, 1:1,000 dilution). NSP12 proteins were detected using an anti-V5 antibody (Invitrogen, 37-7500, 1:1,000 dilution) followed by an Alexa Fluor 488-conjugated secondary antibody (Thermo Fisher, A-11001, 1:1,000 dilution). The nucleus was counterstained with DAPI. Confocal microscopy (Leica TCS SP8 STED, Leica) with an HC PL APO 93×/1.30 motCORR glycerol objective was used to image the samples.

For image-based drug screening, Calu-1 cells, incubated in 96-well plates with drugs and the virus, were blocked with Dulbecco-BSA (0.2%) buffer for 20 min at RT and permeabilized with 0.1% Triton X-100 (Sigma, X-100-500) in Dulbecco-BSA (0.2%). Cells were treated with primary rabbit antibodies against SARS-CoV-2 N protein (Rusanen *et al*, 2021) (1:500 dilution), incubated for 45 min at RT, and washed three times for 15 min with Dulbecco-BSA (0.2%) buffer. Thereafter, the cells were treated with Alexa Fluor goat anti-rabbit IgG 488 (Invitrogen, A11008, 1:1,000 dilution) secondary antibodies. Cells were then incubated for 45 min at RT in the dark and washed three times for 15 min with Dulbecco-BSA (0.2%) buffer. Nuclei were stained with Hoechst (Lifetech, 33342, 1:5,000 dilution). The cells were then imaged with a PerkinElmer Opera Phenix spinning disk confocal microscope using 405nm and 488nm lasers. Images were acquired with Harmony 4.9 software (PerkinElmer).

The image files were processed with SlideBook software (3I Intelligent Imaging Innovations), LAS X (Leica), ImageJ software, and SCT BIAS software.

### Affinity and proximity protein purification

Cells were lysed in ice-cold lysis buffer supplemented with 0.5 mM PMSF and protease inhibitors. For AP-MS, samples were lysed in 3 ml of ice-cold lysis buffer (0.5% IGEPAL, 50 mM HEPES (pH 8.0), 150 mM NaCl, 50 mM NaF, 1.5 mM $NaVO_3$, 5 mM EDTA, 0.5 mM PMSF, and protease inhibitors (Sigma-Aldrich)).

For BioID-MS approach, cell pellets were thawed in 3 ml of ice-cold lysis buffer (0.5% IGEPAL, 50 mM HEPES (pH 8.0), 150 mM NaCl, 50 mM NaF, 1.5 mM $NaVO_3$, 5 mM EDTA, 0.1% SDS, 0.5 mM PMSF, and protease inhibitors), and lysates were sonicated and treated with Benzonase® Nuclease (Santa Cruz Biotechnology, sc-202391).

Cleared lysate was obtained by centrifugation, and the lysate was subjected to a one-step purification via Strep-Tactin® Sepharose® resin (IBA). The purified protein complexes were reduced, alkylated, and digested to peptides for MS analysis. A detailed description of the method used here can be found in a previous protocol (Liu *et al*, 2020).

### Liquid chromatography–mass spectrometry

The analysis was performed on a Q Exactive™ Hybrid Quadrupole-Orbitrap Mass Spectrometer (Thermo Fisher) using Xcalibur version 2.0.7 SP1 (Thermo Fisher) coupled with an EASY-nLC 1000- system via an electrospray ionization sprayer (Thermo Fisher). For each sample, three biological replicates were used, and a 4 μl peptide sample was loaded for each analysis. Peptides were eluted and separated with a C18-packed pre-column and an analytical column, using a 60 min buffer gradient from 5 to 35% buffer B (buffer B: 0.1% formic acid in 98% acetonitrile and 2% HPLC-grade water), followed by a 5-min gradient from 35 to 80% buffer B, and a 10-min gradient from 80 to 100% buffer B at a flow rate of 300 nl/min (buffer A: 0.1% formic acid in 2% acetonitrile and 98% HPLC-grade water). Peptides analysis was performed in a data-dependent acquisition mode using FTMS full scan (200–2,000 *m/z*) resolution of 70,000 and higher-energy collision dissociation (HCD) scan of the top 20 most abundant ions. For protein identification, Thermo .RAW files were searched against selected human plus coronavirus entries in the UniProtKB/SwissProt database (http://www.uniprot. org/) with Sequest search engine with 15 ppm MS1 tolerance, and 0.05 Da fragment mass tolerance. Carbamidomethylation of cysteine was defined as a static modification, and oxidation of methionine and biotinylation of lysine and N-termini were defined as variable modifications. All reported data were based on high-confidence peptides assigned in Proteome Discoverer (Thermo Fisher) with a 1% false discovery rate (FDR) by Percolator.

For phosphoproteomic analysis, LC-MS analysis was performed as before, except was a 120-min linear gradient as the peptide separation gradient. The output data files were processed with MaxQuant version 1.6.0.16 (Cox & Mann, 2008) using the Andromeda search engine (Cox *et al*, 2011) including a dynamic modification of 79.99 Da on serine, threonine, and tyrosine residues. The results were filtered to a maximum false discovery rate (FDR) of 0.05. The processed data were analyzed manually and filtered based on localization probability with a cutoff value of 0.75.

### Identification of the high-confidence interactions

The web tool (http://proteomics.fi/) incorporated with Significance Analysis of INTeractome (SAINT) express version 3.6.0 (Choi *et al*, 2011, 2012) was used as a statistical approach for identification of specific high-confidence interactions from AP-MS and BioID-MS data. 24 GFP control runs (12 N/C-terminal MAC-GFP, 6 C-terminal MAC-GFP with myristoylation signal sequence, and 6 C-terminal MAC-GFP with nuclear localization signal sequence) were used as control counts for each hit. High-confidence interactions (HCIs) were defined by an estimated protein-level Bayesian FDR (BFDR) of ≤ 0.01. Furthermore, we used the CRAPome database (Mellacheruvu *et al*, 2013) with a cutoff frequency of ≥ 20% (≥ 82), except for an average spectral count fold change ≥ 3 to remove the possible false-positive hits.

### Construction of the reference datasets

Human interactome was built by combining unique interactions obtained from Human Cell Map (Go *et al*, 2021), BioPlex (Interactions with probability over 0.95) (Huttlin *et al*, 2021), BioGRID (experimentally detected interactions only) (Oughtred *et al*, 2021), HuRI (Luck *et al*, 2020), PINA2 (Cowley *et al*, 2012), STRING (STRING score > 0.9) (Szklarczyk *et al*, 2019), and IntAct

(experimentally validated physical interactions only) (Kerrien *et al*, 2012). Known interaction pairs for host–receptor baits were retrieved from combined human interactome. Known interactions for viral baits were collected from four recent publications (Gordon *et al*, 2020b; preprint: Laurent *et al*, 2020; preprint: Samavarchi-Tehrani *et al*, 2020; preprint: Stukalov *et al*, 2020). Filters were applied to remove duplicate interaction records, self-interactions, and interactions with bait proteins of other types of coronavirus or organisms.

The bait protein expression profiles of tissues and organs were obtained from the Human Protein Atlas (http://www.proteinatlas.org) (Uhlén *et al*, 2015) that summarizes expression per gene in 62 tissues.

### MS-microscopy analyses

For any bait of interest, the averaged peptide-spectrum match (PSM) values of each prey protein was calculated and uploaded to the web tool (http://proteomics.fi/) to calculate their subceullar distribution.

### Co-Immunoprecipitation

To validate the interaction pairs by Co-IP, HEK293 cells ($5 \times 10^5$ per well) in 6-well plate were co-transfected with Strep-HA-tagged (500 ng) prey and V5-tagged (500 ng) bait constructs using Fugene 6 transfection reagent (Promega). After 24 h of transfection, cells were rinsed with ice-cold PBS and lysed with 1 ml HENN lysis buffer per well (50 mM HEPES pH8.0, 5 mM EDTA, 150 mM NaCl, 50 mM NaF, 0.5% IGEPAL, 1 mM DTT, 1 mM PMSF, 1.5 mM $Na_3VO_4$, 1 × Protease inhibitor cocktail) on ice. The cell lysate was collected, and a clear supernatant was obtained by centrifugation (16,000 *g*, 20 min, 4°C). 30 µl of Strep-Tactin® Sepharose® resin (50% suspension, IBA Lifesciences GmbH) was washed in a micro-centrifuge tube twice with 200 µl HENN lysis buffer (4,000 *g*, 1 min, 4°C). The clear lysate was added to the pre-washed Strep-Tactin beads and incubated for 1 h on a rotation wheel at 4°C. After incubation, the beads were collected by centrifugation and washed three times with 1 ml HENN lysis buffer (4,000 *g*, 30 s, 4°C). After the last wash, 60 µl of 2 × Laemmli sample buffer (Bio-Rad, 1610737) was added directly to the beads and boiled at 95°C for 5 min. Samples were later used for immunodetection via dot-blot.

### Dot-Blot

The Bio-Dot® Microfiltration System (Bio-Rad, 1703938) was assembled according to the manufacturer's instructions. The nitrocellulose membrane was pre-washed with TBS to hydrate the membrane. Ten microliters of sample was spotted onto the nitrocellulose membrane in the center of the well and drained under vacuum pressure. Nonspecific sites were blocked with 5% fat-free milk in TBS-T (0.05% Tween-20 in TBS) for 60 min at RT with gentle shaking. The membrane was then incubated with primary antibody in TBS-T (mouse anti-V5 with a1:5,000 dilution) overnight at 4°C. The membrane was washed three times for 10 min with TBS-T followed by incubation with secondary antibody conjugated with HRP (goat anti-mouse IgG conjugated with horseradish peroxidase with a 1:2,000 dilution) for 60 min at RT with gentle shaking. The membrane was washed three times for 10 min with TBS-T followed by one additional wash with TBS on a shaker. Amersham™ ECL™ Prime (Cytiva) solution was added to the membrane and incubated for 5 min prior to imaging the blot using iBright Imaging Systems

(Thermo Fisher). The same membrane was then stripped by incubating with Restore Plus Stripping buffer (Thermo Fisher) for 15 min and was re-blocked with 5% fat-free milk in TBS-T for 60 min at RT with gentle shaking. The membrane was then incubated with the other primary antibody in TBS-T (mouse anti-HA with a 1:2,000 dilution) overnight at 4°C for different detections.

### Phosphoproteomic analysis

The Flp-In™ 293 T-REx cells and the generated viral stable NSP3-expressing cells were collected and lysed with 8 M Urea buffer containing phosphatase and protease inhibitors cocktail (Sigma-Aldrich, P2745 and P8340) on ice. The cell debris was cleared by centrifugation at 16,000 *g* for 10 min. The protein concertation was determined using a BCA protein assay kit (Pierce, Thermo Fisher). Equal amounts of protein were obtained for all samples and subjected to reduction with 5 mM Tris(2-carboxyethyl)phosphine (TCEP; Sigma-Aldrich), and alkylation with 10 mM iodoacetamide (IAA; Sigma-Aldrich). Protein samples were diluted 4-fold (to less than 2 M urea) with ammonium bicarbonate (AMBIC; Sigma-Aldrich, 213-911-5) before trypsin digestion. The peptide samples were desalted with C18 Macrospin columns (Nest Group). Phospho-peptide enrichment was performed using immobilized metal ion affinity chromatography with titanium (IV) ions ($Ti^{4+}$-IMAC). The IMAC material was prepared and processed as described (Zhou *et al*, 2013). Briefly, $Ti^{4+}$-IMAC beads were loaded onto GELoader tips (Thermo Fisher) and conditioned with 50 µl of conditioning buffer (50% $CH_3CN$, 6% trifluoroacetic acid (TFA)) by centrifuging at 150 *g*. The digested peptide samples were dissolved in the loading buffer (80% $CH_3CN$, 6% TFA) and added into the spin tips with centrifugation at 150 *g*. The column was then washed with 50 µl of wash buffer 1 (50% $CH_3CN$, 0.5% TFA, 200 mM NaCl), followed by 50 µl of wash buffer 2 (50% $CH_3CN$, 0.1% TFA). Bound phospho-peptides were finally eluted with 10% ammonia, followed by a second elution with elution buffer (80% $CH_3CN$, 2% FA). Elution was then dried in a vacuum centrifuge and reconstituted to a final volume of 15 µl in 0.1% TFA and 1% $CH_3CN$ for mass spectrometry analysis.

### Gene Ontology (GO) enrichment analysis and bait–bait correlation analysis

The statistical method SAINT (Choi *et al*, 2011) for probabilistically scoring PPI data was used to define HCIs (Bayesian FDR of ≤ 0.01). All the high-confidence interactors obtained from both AP-MS and BioID-MS were subjected to KEGG database (https://www.genome.jp/kegg/) (Kanehisa *et al*, 2015) and Reactome pathway-based enrichment analysis (Fabregat *et al*, 2017). GO term fusion was used, and only enriched terms with *P* values ≤ 0.01 were displayed.

The SAINT processed file with quantitative information on bait–prey interactions was uploaded to the web tool (https://prohits-viz.org/) (Knight *et al*, 2017) to perform correlation analysis and generate detailed bait–bait comparisons.

### Network and interaction map

Protein interaction data were imported into Cytoscape 3.8 (Shannon *et al*, 2003) for PPI network visualization. The relevant protein interaction network maps are available at https://www.ndexbio.org/.

### Scoring prey proteins of the virus–host interactome

The protein connectivity of proposed huh protein in human interactome was summed based on the unique interaction found in the combined human interactome.

To calculate the cumulative frequency of proposed huh protein in SARS-CoV-2 interactome, we incorporated a total of six datasets, including our virus–host AP-MS and BioID-MS datasets and four other proteomic datasets to build a complete SARS-CoV-2 interactome (Gordon *et al*, 2020b; preprint: Laurent *et al*, 2020; preprint: Samavarchi-Tehrani *et al*, 2020; preprint: Stukalov *et al*, 2020). A list of 4477 proteins was compiled (Dataset EV11). Frequency of prey protein in the SARS-CoV-2 interactome was counted and adjusted according to its occurrence of the datasets.

$$Adjust\,frequency\,of\,occurrence = \frac{\sum_{i=1}^{n} \text{Frquency counts in all database}}{6} \times \sum_{l=1}^{m} Occurance\,in\,datasets$$

To be precise, the cumulative frequency of a prey protein was calculated by summing up the frequency of occurrence ($n$) within each dataset. The cumulative frequency was divided by the total number of datasets (6) and then multiplied by the appearance within database ($m$, maximum is 6) to obtain the adjusted frequency of occurrence. The adjusted score represents the overall frequency of the occurrence of a protein: The lower the score is, the less connects of the protein receives.

### Protein preparation for molecular docking and druggability assessment

The PDB IDs of potential target proteins detected during interaction analysis were collected based on the structural information available in UniProt (https://www.uniprot.org/). Crystal structures with bound small molecules or cofactors were prioritized, and each target protein was subjected to a tailored KNIME workflow for protein structure preparation (Schrödinger Release 2020-3: Schrödinger KNIME Extensions, Schrödinger, LLC, New York, NY, 2020). The workflow consisted of four steps: (a) structure retrieval and conversion to Maestro format, (b) protein preparation, (c) structural alignment, and (d) exporting structures to a Maestro project for visual inspection. The protein preparation (step b) involved the addition of hydrogens, bond order assignment, and deletion of water molecules beyond 5 Å from heteroatoms. Next, the hydrogen bonding network was optimized, and structures were minimized using the OPLS3e forcefield until the heavy atom RMSD between iterations converged to 0.3 Å. The first input crystal structure for each target was chosen as the template for the structural alignment (step c). Visual inspection and structure validation were performed to aid the selection of the most suitable docking receptors. For those receptors, a second protein preparation step was performed within Schrödinger Maestro (Schrödinger Release 2020-3: Maestro, Schrödinger, LLC, New York, NY, 2020) to add missing side chains with Prime and remove solvent molecules, ions, and crystallization additives prior to hydrogen bond network optimization. After that, ligands were split from the structure and a druggability assessment with SiteMap was carried out to aid the final target receptor selection (Schrödinger Release 2020-3: SiteMap, Schrödinger, LLC, New York, NY, 2020) (Halgren, 2007, 2009). Up to 5 top-ranked potential receptor binding sites were identified with the more restrictive definition of

hydrophobicity, with each requiring at least 15 site points and being cropped at 4 Å from the nearest site point. Sites overlapping with the crystallographic ligand-binding sites were further evaluated. The PDB IDs of the chosen docking receptors and the corresponding results of the druggability analysis are presented in Dataset EV12.

### Database preparation for molecular docking

Drug molecules were collected from DrugBank (Knox *et al*, 2011), ChEMBL (Bento *et al*, 2014), the NPC browser (Huang, Southall, *et al*, 2011), and the SciFinder database (scifinder.cas.org). The resulting collection was pre-processed manually to retain largely typical drug-like molecules by removing inorganic compounds, common substrates/cofactors such as ATP and NADPH, and applying a molecular weight cutoff of 700 g/mol. From the remaining 5,518 unique compounds, a Phase database was prepared using the LigPrep module of Schrödinger Suite 2020-3 (Schrödinger Release 2020-3: Phase, Schrödinger, LLC, New York, NY, 2020) with default parameters and retaining the specified chirality where applicable.

### Docking

For each docking receptor, a grid (docking parameter file) was generated with the Glide module of Schrödinger. The binding site center was defined based on the centroid of the crystallographic ligand. For the kinase proteins, knowledge-based hydrogen bond constraints were included for the so-called hinge-region amide to enhance the convergence of these virtual screenings. The grids were used for the Glide virtual screening workflow, where an initial screening was performed by Glide high-throughput virtual screening (HTVS) docking, followed by docking the 20% top hits with standard precision (SP) mode (Schrödinger Release 2020-3: Glide, Schrödinger, LLC, New York, NY, 2020). The top-scoring 200 ligands in the SP results were docked with extra precision (XP) mode in a final step. At all docking stages, the ligand van der Waals potentials were softened with a factor of 0.6 up to a partial charge cutoff of 0.15 e. The final hit selection was based on visual inspection of the docking poses of top-ranked ligands from SP and XP docking to identify favorable ligand geometries and interactions in the protein-binding site. For each protein target, up to eight final repurposing candidates were selected and are summarized in Dataset EV12.

### Image-based drug screen with SARS-CoV-2

Drugs dissolved in DMSO or water according to the manufacturers' protocols were dispensed into the wells of CellCarrier-96 Ultra Microplates (PerkinElmer, 6055302) in six different concentrations using acoustic Echo 550 liquid handler (Labcyte) (FIMM High Throughput Biomedicine Unit, FIMM, HiLIFE, University of Helsinki, Finland). Calu-1 cells were seeded into pre-drugged plates in density 10,000 cells/well in a standard cell culture medium. Thereafter, the cells were infected with SARS-CoV-2 at a multiplicity of infection (MOI) of 10. Mock-infected cells (non-virus-infected and non-drug-treated) exposed to DMSO were used as a control for the virus (virus-infected cells) in the presence of DMSO inhibition and cell viability assessment. Non-drug-treated cells infected with the virus served as an infection control. After incubation for 48 h at 37°C, cells were fixed with 4% PFA. Immunofluorescence staining was performed on fixed cells, and images were acquired using the Opera Phenix HT microscope (see the section for

Immunofluorescence) (FIMM High Throughput Biomedicine Unit, FIMM, HiLIFE, University of Helsinki, Finland).

### Drug response quantification

Cells in the acquired images were segmented by expanding the nuclear region (based on nuclear staining) and classified to N protein-positive and N protein-negative populations using the intensity and morphology features with a training set of 150 cells from mock-infected and virus-infected controls, respectively. A four-parameter (4PL) logistic regression model was used to fit dose–response data points to calculate the half-maximal inhibitory concentrations (IC50) and the area under the curve (AUC) for each drug by using the virus-infected control values (without the presence of drugs) as a baseline (Bailer, 1988; Jaki & Wolfsegger, 2009). Curve fitting and AUC were used to quantify the drug-induced inhibition of virus infection (count of cells positive for N protein) and overall cytotoxicity (total nuclei count) with the Prism 4.2 software (GraphPad Software, San Diego, California USA). A larger AUC value indicates increased effect of the evaluated drug. We considered drugs with an AUC value (count of cells positive for N protein) > 100 with a positive curvature as an effective antiviral compound.

### Compound treatment for the immunoprecipitation assay

For the cell-based immunoprecipitation, compounds were added to the culture medium of HEK293 cells for 24 h before harvesting. Strep-HA-tagged GFP with V5-tagged LacZ construct was used as a negative control. The compounds BMS-863233 (XL413, CAS: 1169562-71-3) and methotrexate (CL-14377, CAS: 59-05-2) were purchased from Selleck Chemicals. Cells were, each for 24 h, untreated or treated with 5 μM and 50 μM BMS-863233 (dissolved in water) or 1 μM and 10 μM methotrexate (dissolved in DMSO). Each condition was repeated three times to generate the average trend.

## Data availability

The datasets produced in this study are available in the following databases:

MS data: MassIVE (https://massive.ucsd.edu/) with web access MSV000087035.

Protein–protein interaction data: IMEx consortium (http://www.imexconsortium.org).

The interaction network presenting by Cytoscape: NDEx (https://ndexbio.org/).

Image-based drug screening data and images: Zenodo.org (10.5281/zenodo.5534941).

Expanded View for this article is available online.

## Acknowledgements

We thank Sini Miettinen (Proteomics Unit, Institute of Biotechnology & HiLIFE) for technical assistance of mass spectrometry analysis, as well as prof. Nevan Krogan (University of California-San Francisco, USA) for several SARS-CoV-2-related ORF cDNA plasmid clones. Imaging was performed at the Light Microscopy Unit (LMU), Institute of Biotechnology, supported by HiLIFE and Biocenter Finland. NanoString analysis was carried out at the Genomics Unit at Institute of Biotechnology. The authors wish to acknowledge CSC-IT Center for Science Ltd. Finland for computational resources. The FIMM High Throughput Biomedicine Unit (FIMM-HTB), especially Laura Turunen, is acknowledged for preparing and designing drug plates and the layouts, and the FIMM High Content Imaging and Analysis Unit (FIMM-HCA) for HC-imaging and analysis (HiLIFE, University of Helsinki and Biocenter Finland, Helsinki, Finland). We thank Prof. Olli Vapalahti, Prof. Olli Kallioniemi, and PhD Suvi Kuivanen (University of Helsinki, Finland, and the iCOIN consortium) for sharing facilities, protocols, and expertise. This work was supported by the Academy of Finland (#288475 and CoVIDD #336470 for MV), Academy of Finland (#336470), and Jane and Aatos Erkko Foundation (to MKV). Academy of Finland (VP and MB: iCOIN-336496 and 308613, AH: FIRI2020-337036 for FIMM-HCA). The authors thank the support from NanoString Technologies, Inc. "Science Never Stops" Grant (to XL). The authors thank Biocenter Finland/DDCB for financial support.

## Author contributions

MV and XL designed and conceptualized the study. XL, SH, KS, TÖ, LG, SK, and MV generated proteomic data and performed data analysis. TR and MKV conducted experiments to check the subcellular localization of viral ORFs. TL, IP, AKT, and AP performed drug repurposing virtual screening. MB, AH, and VP performed image-based drug screening and data analysis. MV and XL wrote the manuscript with inputs from all authors.

## Conflict of interest

The authors declare that they have no conflict of interest.

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
