## [Review Process File · Molecular Systems Biology]

SARS-CoV-2 -host proteome interactions for antiviral drug discovery

Xiaonan Liu, Sini Huuskonen, Tuomo Laitinen, Taras Redchuk, Mariia Bogacheva, Kari Salokas, Ina Pöhner, Tiina Öhman, Arun Tonduru, Antti Hassinen, Lisa Gawriyski, Salla Keskitalo, Maria Vartiainen, Vilja Pietiäinen, Antti Poso, and Markku Varjosalo
DOI: 10.15252/msb.202110396

Corresponding author(s): Markku Varjosalo (markku.varjosalo@helsinki.fi)

Review Timeline:

Submission Date:	12th Apr 21
Editorial Decision:	14th May 21
Revision Received:	26th Aug 21
Editorial Decision:	13th Sep 21
Revision Received:	1st Oct 21
Accepted:	4th Oct 21

Editor: Maria Polychronidou

Transaction Report:

Thank you again for submitting your work to Molecular Systems Biology. We have now heard back from the three referees who agreed to evaluate your study. As you will see below, the reviewers raise substantial concerns, which unfortunately preclude the publication of the study in its current form.

The reviewers point out that the overall conceptual novelty remains limited given that SARS-CoV-2-human protein-protein interactions and viral protein subcellular localization have been investigated in previous studies. Reviewer #2 specifically mentioned that as it stands the impact of the study seems limited and that in absence of follow up experimental validations several of the main conclusions remain rather tentative. During our cross-commenting process, in which the reviewers are given the opportunity to comment on each other's reports, reviewer #3 mentioned that they agree with the comments of reviewer #2 regarding the lack of validation of the presented predictions and pointed out that in their opinion follow up experimental validations (of some) of the predicted therapeutic interventions would be required.

Given that all three reviewers acknowledge that the study is comprehensive and could serve as a potentially useful resource despite the somewhat limited conceptual novelty, we have decided to offer you a chance to address the issues raised in a major revision.

Without repeating all the points listed below, the more fundamental issues that need to be convincingly addressed are the following:

- Further experimental evidence should be provided to better support the proposed role of (some of) the proteins in viral entry/infection.
- As reviewer #2 indicates, "validating [at least some of] the identified molecules (those that do and do not inhibit interactions) for antiviral efficacy or validating [at least some of] the preys as dependency factors via a genetic screen" would be required to increase the conclusiveness and overall impact of the study. As mentioned above, reviewer #3 agreed with the importance of including analyses along those lines.
- Reviewer #2 also raises important points related to the data analysis, which need to be carefully addressed.

All issues raised by the reviewers need to be satisfactorily addressed. As you may already know, our editorial policy allows in principle a single round of major revision, so it is essential to provide responses to the reviewers' comments that are as complete as possible. I understand that the required revisions are substantive. Please feel free to contact me in case you would like to discuss in further detail any of the issues raised or if you would like to share your revision plan with me.

On a more editorial level, we would ask you to address the following issues:

REFEREE REPORTS

Reviewer #1:

This is another database of CoV2-host protein interactions that are being presented. While there have been other such studies in the literature in the past year, there has been very little overlap in the host proteins identified. This work is a bit more comprehensive and uses cells expressing the ACE2 receptor for transfecting various ORFs, nsps and structural proteins. Some of its findings appear to overlap with existing literature thus increasing confidence, and other findings such as the change in nuclear actin dynamics and the disruption of the rods/rings phenomena induced by ribavarin appear to be novel. Given this, although it does not have much hypothesis or testing/validation of conjectures, I favor publication as it does provide some novel host targets that might be of interest for COVID biology and therapeutic manipulation.

Reviewer #2:

In the manuscript, "SARS-CoV-2 -host proteome interactions for antiviral drug discovery", Liu et al. perform affinity purification mass spectrometry of 29 SARS-CoV-2 proteins and 18 human proteins. They also include localization analysis of SARS-CoV-2 proteins as well as staining of actin upon overexpression of SARS-CoV-2 proteins. They functionally annotate preys and cluster baits based on common prey function. Lastly, they identify a list of drugs that could potentially inhibit virus-human protein-protein interactions. One can easily appreciate the amount of work that went into this manuscript, and for that I applaud the authors. However, much of this work lacks novelty. SARS-CoV-2-human protein-protein interactions have been investigated by several other groups as has subcellular localization of viral proteins. Moreover, the manuscript lacks experimental validation of the hypothetical mechanisms extracted from the dataset. Lastly, the authors should consider either validating [at least some of] the identified molecules (those that do and do not inhibit interactions) for antiviral efficacy or validating [at least some of] the preys as dependency factors via a genetic screen.

Additional comments:

- 0.1% recovery of host PPIs is very low (i.e. 99.9% novel). Typically one would expect to discover on the order of 80% novel interactions. The authors should elaborate as to why their study picks up so few known interactions. Could this be attributed to how they scored high-confidence interactions? The authors should evaluate different methods of thresholding their dataset.
- The statement, "Moreover, more than 50% of the prey proteins were detected to have interactions with more than one viral baits" should be investigated further. This is a bold statement. The Gordon et al. Nature (2020a) study found no such redundancy. Viral proteins are generally thought to be highly specialized, with lower overall redundancy. Indeed, viruses must achieve much with a relatively small genome. Perhaps the authors could assess the confidence (e.g. abundance in IP, interaction score, etc) of interactions that appear to be interacting with multiple baits in order to convince the reader of this statement.
- Could the enrichment for Calnexin pathways simply be an indication that the baits that are expressed are inappropriately folded or perhaps due to overexpression are activating mechanisms of protein quality control? The authors should consider this possibility if they choose to highlight this pathway/process in their manuscript.
- This statement "Indicating SARS-CoV-2 infection could perturb the cell cycle" should be cited with Bouhaddou et al., Cell (2020) (and maybe others) that have found SARS-CoV-2 to provoke cell cycle arrest.
- Subcellular localization analysis fails to cite Gordon et al., Science (2020). Furthermore, the authors could compare their results to Gordon et al., 2020 and Zhang, Cruz-cosme et al., 2020 as a supplemental analysis.
- It would be nice if the authors could do an analysis to prove that viruses target highly-connected hub proteins using either a statistical or network analysis.
- Regarding the statement: "Around 10% (7/59) of the proposed drugs have already been introduced to clinical studies to limit complications for COVID-19 patients" -- How does this compare to what one would expect by chance? For instance, if you chose a random set of 59 drugs, would you find your PPI-driven approach enriches for drugs entering COVID clinical trials?
- The authors look for drugs that have the potential to modulate virus-human protein-protein interactions. Either the authors should test these drugs to show their ability to modulate viral replication or they should focus on targeting interactions with human proteins previously shown to be dependency factors for the virus. There are previous genome-wide screens for SARS-CoV-2 infection that could be used in this regard. In general, the drug repurposing section of the manuscript could use a clearer motivation embedded in experimental evidence for why certain virus-human protein-protein interactions should be perturbed.
- Scoring of high-confidence protein-protein interactions could use some improvement. Either the authors should explain their

analysis in greater detail (Methods did not contain enough detail) and/or use more popular approaches such as SAINT, compPASS, MiST, or a combination of them.

- Figure 5a missing labels for study origin.

Reviewer #3:

In this work, Liu and colleagues generated an impressive number of stably transfected cell lines to perform a not less impressive number of AP-MS, BioID, and immunofluorescence experiments. This provides insight into the biology of SARS-CoV-2 and offers suggestions potential new therapeutic avenues - although the authors do not validate any of these.

Although, similar studies have been published or are available as pre-prints, this work constitutes an important addition to the number of systematic studies of SARS-CoV-2 biology. Overall, the study is well designed and the language is understandable, but could benefit from a revision from a native speaker. My major suggestions are that the authors put the study better in the context of current knowledge and that they facilitate the access to the data generated in this study.

In this work, Liu and colleagues generated cell lines stably expressing all 29 SARS-CoV-2 proteins and 18 host proteins that have been recognized to be important for viral entry and replication, fused to a MAC-tag system. This allowed performing both AP-MS and BioID from the same constructs. The authors further perform immunofluorescence experiments with all viral proteins (which only failed for ORF9b) to detect their subcellular localization, which together with the MS data can give hints on the function of the viral proteins (e.g., this work shows an involvement of NSP3 in perinuclear actin polymerization). The authors finally use this knowledge together with virtual drug screening to propose potential therapies - some of which have been previously suggested by others.

The authors acknowledge the limitations of expressing one viral protein at a time, outside of an infection context, and the possibility of missed interactions. Even though similar studies have been published or are available as pre-prints, this work constitutes an impressive effort. Given the general low overlap between similar studies, another such effort allows us to start mapping the core interactions of viral proteins, as well as accessory interactions that might be cell type specific or artifacts of sample processing.

The study is overall well designed and I have only some minor comments that I believe could improve the current manuscript:

1. The language is clear, but would benefit from a careful revision from a native speaker, as there are some grammatical errors throughout the manuscript.
2. Given the hairball nature of the interaction network (and the difficult in navigating it in a PDF, due to the large number of vector elements), it would be good if the authors made available a cytoscape file or a web-based platform that would allow queries for specific proteins and filtering for their interactors.
3. The authors could include an analysis of the overlap between the AP-MS and BioID results. What are the commonalities and what are the differences? Perhaps, this should be even expanded to include all currently published results of these experiments, to really get at the core of protein interactions from viral proteins.
4. Figure 2 would benefit from the inclusion of the information of which compartment the authors would assign each protein, based on the imaging. The authors could also compare their MS-microscopy results (Supplementary Figure 12) with the actual microscopy. Do all proteins agree?
5. The section "Functional characteristics of hub proteins involving in critical signaling pathways" reads mostly like a review of the literature and I feel that it falls short of putting the results in context. While figure 4 points out the frequency at which these proteins were preys in the experiments from the authors, they do not link to specific viral proteins. The authors should consider rewriting this section to link specific viral proteins to the host processes highlighted.
6. The methods section lacks most aspects of how data analysis was performed. A few examples: how the MS-microscopy was carried out?, how the correlations in Figure 3 were performed?. It would be important to go through all figures/analyses and include these details in the methods.
7. It would be good if supplementary tables 1 and 2 were more harmonized. For example, Supplementary table 2 lacks the information if host proteins were tagged in the C- or N-terminus.

Point-by-point response to the referees

We want to thank the reviewers for their thorough reading and their very constructive suggestions that were very helpful to revise and improve our manuscript. In particular, Reviewers 2 and 3 commented on validating some of the identified molecules for antiviral efficacy, which we have addressed in the revised version. Specifically, we were validating 190 interaction pairs via co-IP and 10 drug candidates via viral infectivity assay. Two of the promising agents, methotrexate and BMS-863233, was further investigated for interfering the protein interaction tests. Our detailed point by point response follows below.

Reviewer #1:

This is another database of CoV2-host protein interactions that are being presented. While there have been other such studies in the literature in the past year, there has been very little overlap in the host proteins identified. This work is a bit more comprehensive and uses cells expressing the ACE2 receptor for transfecting various ORFs, nsps and structural proteins. Some of it's findings appear to overlap with existing literature thus increasing confidence, and other findings such as the change in nuclear actin dynamics and the disruption of the rods/rings phenomena induced by ribavarin appear to be novel. Given this, although it does not have much hypothesis or testing/validation of conjectures, I favor publication as it does provide some novel host targets that might be of interest for COVID biology and therapeutic manipulation.

We thank the Reviewer for the concise summary. We appreciate the time and effort that Reviewer has dedicated to providing her/his supportive comments on our manuscript. Although this reviewer was already in favor of publication, we have now added significant validation of the detected interactions (>100 interactors), as well as live virus-based inhibition assays for several of the proposed compounds identified using our pipeline. We believe that now the manuscript is of even further interest for COVID biology and it's therapeutics.

Reviewer #2:

In the manuscript, "SARS-CoV-2 -host proteome interactions for antiviral drug discovery", Liu et al. perform affinity purification mass spectrometry of 29 SARS-CoV-2 proteins and 18 human proteins. They also include localization analysis of SARS-CoV-2 proteins as well as staining of actin upon overexpression of SARS-CoV-2 proteins. They functionally annotate preys and cluster baits based on common prey function. Lastly, they identify a list of drugs that could potentially inhibit virus-human protein-protein interactions. One can easily appreciate the amount of work that went into this manuscript, and for that I applaud the authors. However, much of this work lacks novelty. SARS-CoV-2-human protein-protein interactions have been investigated by several other groups as has subcellular localization of viral proteins. Moreover, the manuscript lacks experimental validation of the hypothetical mechanisms extracted from the dataset. Lastly, the authors should consider either validating [at least some of] the identified molecules (those that do and do not inhibit interactions) for antiviral efficacy or validating [at least some of] the preys as dependency factors via a genetic screen.

We thank the Reviewer for her/his very constructive suggestions. We have included the additional data (interaction pairs validation and drug screening) to demonstrate that our large-scale analysis is a useful resource.

Additional comments:

- 0.1% recovery of host PPIs is very low (i.e. 99.9% novel). Typically one would expect to discover on the order of 80% novel interactions. The authors should elaborate as to why their study picks up so few known interactions. Could this be attributed to how they scored high-confidence interactions? The authors should evaluate different methods of thresholding their dataset.

We thank the Reviewer to point out this very important issue about known interactions. Therefore, we have further integrated protein interaction data from seven major databases, including human cell map (Go, Knight et al. 2021), BioPlex (Huttlin, Bruckner et al. 2021), BioGRID (Oughtred, Rust et al. 2021), HuRI (Luck, Kim et al. 2020), PINA2 (Cowley, Pinese et al. 2012), STRING (Szklarczyk, Gable et al. 2019), and IntAct (Kerrien, Aranda et al. 2012). In total, 2465 known interactions that relate to 18 host bait proteins were retrieved (Fig. EV1). However, less than half of these interactions were constantly collected by two or more databases, indicating these interactions were detected by different approaches. In total, 93 interactions were also identified in this study (93/4362). This equals to 2.1% of known interaction for our dataset. We have updated the corresponding part of the text (**Result section “proteome interaction data validation”, first paragraph, page 5**), and **Fig EV1**.

Regarding the ratio of known interaction, with AP-MS, especially for identifying the stable protein complexes, it is commons to discover on the order of 50% novel interactions. However, when parallel combining with BioID-MS approach, it is common to have less than 5% known interaction. Since BioID-MS can identify transient protein interactions and proximal protein interactions, this is actually expect difference). This is especially true for less studied proteins/protein complexes. For example, earlier research project using AP-MS and BioID-MS to profile centrosome-cilium interface (PMID: 26638075), researchers identified 7092 high confidence interaction including 213 known interaction (3% of known interactions). Fewer known interactions also indicate the necessity of our study.

- The statement, "Moreover, more than 50% of the prey proteins were detected to have interactions with more than one viral baits" should be investigated further. This is a bold statement. The Gordon et al. Nature (2020a) study found no such redundancy. Viral proteins are generally thought to be highly specialized, with lower overall redundancy. Indeed, viruses must achieve much with a relatively small genome. Perhaps the authors could assess the confidence (e.g. abundance in IP, interaction score, etc) of interactions that appear to be interacting with multiple baits in order to convince the reader of this statement.

We thank the Reviewer to point out this potential misleading sentence. Accordingly, we revised the text to emphasize two possible explanations 1), this implies that viral ORFs may appear in the same sub-cellular region and, therefore, similar proximal proteins were detected. 2), it may suggest the virus targeting the same host factor in redundant ways, and a further investigation is needed. We have updated the corresponding part of the text (**Result section “proteome interaction data validation”, third paragraph, page 6**).

...This implies that viral ORFs may appear in the same subcellular region; therefore similar proximal proteins were detected. Alternatively, it may also suggest that the virus targets the same host factor in redundant ways, and further investigation is needed.

Furthermore, we conducted co-IP and dot blotting experiments (**Fig EV3 and Dataset EV6**) to validate some of these interactions regarding the possibility of multiple viral baits using the same proteins for processing. For example, multiple viral baits interact with Unconventional myosin-Ib (MYO1B). We validated 10 interaction pairs, and 8 of them were positive via co-IP. (**Fig EV3 and Dataset EV 6**). The possible explanations of low redundancy of the Gordon et al. Nature (2020a) study could be because 1), we used 5 times more cells (cells collecting from 5x 150-mm cell culture plates were used as one biological replicate) than them (cells collecting from 1x 150-mm cell culture plates were used as one biological replicate) for AP-MS/BioID-MS. It is a more labor-

intensive method but provides us higher coverage of interaction partners. 2), depending on the control samples they used and the statistical approach they applied on the data, the cut-off for the occurrence of the prey protein in the specific database may not be the same as we did. Therefore, profiling the interactome of SARS-CoV-2 with different approaches allows the recovery of distinct interactions and can be considered a work in progress.

- Could the enrichment for Calnexin pathways simply be an indication that the baits that are expressed are inappropriately folded or perhaps due to overexpression are activating mechanisms of protein quality control? The authors should consider this possibility if they choose to highlight this pathway/process in their manuscript.

We thank the Review for pointing this out and we agree with this comment. We have now acknowledged this possibility in the paragraph and added to the Discussion. (**Discussion section, paragraph 4, page 14-15**).

...Thirdly, we highlighted ER-calnexin pathway in our study to address possible relationship with viral processes. However, overexpression of MAC-tagged bait protein could also induce the unfolded protein response, which results in the up-regulation of genes encoding ER-resident chaperones such as calnexin, calreticulin, and hypoxia-upregulated 1 gene (HYOU1) (Lindholm, Korhonen et al. 2017). Although, these proteins are not often or systematically detected in large-scale interactome studies, further experiments will be needed to rule out the possibility.

- This statement "Indicating SARS-CoV-2 infection could perturb the cell cycle" should be cited with Bouhaddou et al., Cell (2020) (and maybe others) that have found SARS-CoV-2 to provoke cell cycle arrest.

We apologize for the oversight. We have now added citations to several of these (Bouhaddou, Memon et al. 2020, Tutuncuoglu, Cakir et al. 2020).

- Subcellular localization analysis fails to cite Gordon et al., Science (2020). Furthermore, the authors could compare their results to Gordon et al., 2020 and Zhang, Cruz-cosme et al., 2020 as a supplemental analysis.

We thank the Reviewer for the suggestion and apologize for the negligence. We have compared results with previous publications (Gordon, Hiatt et al. 2020, Zhang, Cruz-cosme et al. 2020, Lee, Huang et al. 2021), no obvious difference was observed by light microscopy of the subcellular localization of single ORF expressing in different cell lines (**Dataset EV7**). This data has now been added to the manuscript (**Result section "Subcellular localization of viral bait proteins", page 7**).

- It would be nice if the authors could do an analysis to prove that viruses target highly-connected hub proteins using either a statistical or network analysis.

We thank the Reviewer for the suggestion. To verify those 693 hypothetical hub proteins are essential for viral replication, we designed a simple scoring strategy (**Method part, "Scoring prey proteins of the virus-host interactome", page 20-21**) to score prey protein by incorporating our virus-host AP-MS and BioID-MS datasets and four other proteomic datasets (Gordon, Jang et al. 2020, Laurent, Sofianatos et al. 2020, Samavarchi-Tehrani, Abdouni et al. 2020, Stukalov, Girault et al. 2020). The compiled list yields 4477 proteins (**Dataset EV11**). The adjusted frequency was then applied to the list based on the occurrence of prey protein in different datasets and the frequency of the specific prey in all datasets. Interestingly, although 693 hypothetical hub proteins cover only about one-sixth of all proteins in the list, the majority of these proteins are with high adjusted-frequency (**Appendix Fig. S10**), reflecting their functional essentiality by topological importance in the interaction network. This analysis has been included into the manuscript (**Result section "Viruses target highly-connected and central host proteins involved in critical cellular functions", paragraph 2, page 9**).

- Regarding the statement: "Around 10% (7/59) of the proposed drugs have already been introduced to clinical studies to limit complications for COVID-19 patients" -- How does this compare to what one would expect by chance? For instance, if you chose a random set of 59 drugs, would you find your PPI-driven approach enriches for drugs entering COVID clinical trials?

We thank the Reviewer for this suggestion and considered the proposed comparison against a random baseline. However, the list of COVID clinical trials we used to arrive at this initial statement is biased, since we only took large scale clinical trials registered at <https://clinicaltrials.gov>, <https://www.clinicaltrialsregister.eu> or <https://www.isrctn.com> into account. As registration is not mandatory and those databases mainly cover trials in the US, Europe and the UK, our list of trials is by no means complete. We further did not take published smaller pilot studies into account. Lastly, the reported number of compounds evaluated for use in the context of COVID in patients is changing rapidly, as the preparation of manuscripts describing those studies takes time and additional clinical trials are continuously launched. Furthermore, several of the compounds in clinical trials for COVID are not (directly) used to treat COVID, but, as we stated initially, to limit complications rooted in other conditions, such as diabetes. Thus, we cannot conclude that entry into a clinical trial in the COVID context implies anti-viral activity – however, at the same time, we cannot exclude activity, even if the primary indication and motivation for using a specific drug in a clinical trial is not COVID itself.

For the above-mentioned reasons, we believe that robust statistics to address this question are difficult to achieve and beyond the scope of the current study. We have therefore removed the statement from the manuscript. We believe that the *in vitro* validation of anti-viral activity for selected repurposing candidates added to the revised manuscript presents a better empirical way to support the validity of our approach.

- The authors look for drugs that have the potential to modulate virus-human protein-protein interactions. Either the authors should test these drugs to show their ability to modulate viral replication or they should focus on targeting interactions with human proteins previously shown to be dependency factors for the virus. There are previous genome-wide screens for SARS-CoV-2 infection that could be used in this regard. In general, the drug repurposing section of the manuscript could use a clearer motivation embedded in experimental evidence for why certain virus-human protein-protein interactions should be perturbed.

Thank the Reviewer for these insightful comments. We have used the image-based drug screen assay with infectious SARS-CoV-2 to show antiviral effects of candidate drugs *in vitro* (**Dataset EV 13 and Appendix Fig S11**). In addition, the antiviral activity of methotrexate and BMS-863233 was examined to associate to their inhibitory effects on specific protein-protein interactions. Our results indicated that methotrexate could inhibit viral replication and entry by inhibiting interactions of DDX39B with NSP13, NSP14, S protein and TMPRSS2 (**Fig 6**). However, we did not observe any effects on interactions related to GLO, in the presence of BMS-863233 (**Appendix Fig S12**). We have, accordingly, revised the manuscript to emphasize this point in a section newly added to the Result part (**Result section “Antiviral activities of proposed drug candidates”, page 12-13**).

- Scoring of high-confidence protein-protein interactions could use some improvement. Either the authors should explain their analysis in greater detail (Methods did not contain enough detail) and/or use more popular approaches such as SAINT, compPASS, MiST, or a combination of them.

We are sorry for causing unnecessary confusion. We used web tool (<http://proteomics.fi/>) incorporated with Significance Analysis of INteractome (SAINT) express version 3.6.0 (Choi, Larsen et al. 2011, Choi, Liu et al. 2012) as a statistical approach for identification of specific high-confidence interactions from AP-MS and BioID-MS data. This information was briefly mentioned in the original text. We have edited and expanded the methodology in the manuscript (**Method section, “Identification of the high confidence interactions”, page 18**).

- Figure 5a missing labels for study origin.

We thank the reviewer for noticing this unclarity. We have now changed “Host protein candidate” to read “Proposed hub proteins”.

Reviewer #3:

In this work, Liu and colleagues generated an impressive number of stably transfected cell lines to perform a not less impressive number of AP-MS, BioID, and immunofluorescence experiments. This provides insight into the biology of SARS-CoV-2 and offers suggestions potential new therapeutic avenues - although the authors do not validate any of these.

Although, similar studies have been published or are available as pre-prints, this work constitutes an important addition to the number of systematic studies of SARS-CoV-2 biology. Overall, the study is well designed and the language is understandable, but could benefit from a revision from a native speaker. My major suggestions are that the authors put the study better in the context of current knowledge and that they facilitate the access to the data generated in this study.

In this work, Liu and colleagues generated cell lines stably expressing all 29 SARS-CoV-2 proteins and 18 host proteins that have been recognized to be important for viral entry and replication, fused to a MAC-tag system. This allowed performing both AP-MS and BioID from the same constructs. The authors further perform immunofluorescence experiments with all viral proteins (which only failed for ORF9b) to detect their subcellular localization, which together with the MS data can give hints on the function of the viral proteins (e.g., this work shows an involvement of NSP3 in perinuclear actin polymerization). The authors finally use this knowledge together with virtual drug screening to propose potential therapies - some of which have been previously suggested by others.

The authors acknowledge the limitations of expressing one viral protein at a time, outside of an infection context, and the possibility of missed interactions. Even though similar studies have been published or are available as pre-prints, this work constitutes an impressive effort. Given the general low overlap between similar studies, another such effort allows us to start mapping the core interactions of viral proteins, as well as accessory interactions that might be cell type specific or artifacts of sample processing.

We thank the Reviewer for giving us the opportunity to submit a revision of our manuscript. We have incorporated changes to reflect most of the suggestions provided by the Reviewer.

The study is overall well designed and I have only some minor comments that I believe could improve the current manuscript:

1. The language is clear, but would benefit from a careful revision from a native speaker, as there are some grammatical errors throughout the manuscript.

We thank the Review for this suggestion. The manuscript has been edited by English language editing services. We attached the certification in our submission file.

2. Given the hairball nature of the interaction network (and the difficult in navigating it in a PDF, due to the large number of vector elements), it would be good if the authors made available a cytoscape file or a web-based platform that would allow queries for specific proteins and filtering for their interactors.

We are grateful to the Reviewer for helping us improve the usability of the respective interaction data. All relevant protein interaction map are available now on <https://www.ndexbio.org/> (please

search for user: xioliu to check more details). With “enter query terms” function on NEDx, user could search for specific proteins and relevant information.

3. The authors could include an analysis of the overlap between the AP-MS and BioID results. What are the commonalities and what are the differences? Perhaps, this should be even expanded to include all currently published results of these experiments, to really get at the core of protein interactions from viral proteins.

We want to thank the Reviewer for her/his very constructive suggestions. We incorporated your suggestion by compiling all the interaction data to generate a prey list (4477 unique identifier), which indicates the frequency of each prey. The core protein should be identified by most of the datasets, independent on the cell lines and workflows. The higher frequency the prey is, the more essential the protein should be. Considering the possible bias of each dataset, we adjust the frequency of prey based on the occurrence by the database to generate the adjusted frequency. Interestingly, although 693 hypothetical hub proteins cover only about one-sixth of all proteins in the list, the majority of these proteins are with high adjusted-frequency (**Appendix Fig. S10**), reflecting their functional essentiality by topological importance in the interaction network. This result also emphasizes the necessity by incorporating the host-bait interactome to identify the viral essential protein for replication in host. This analysis has been included into the manuscript (Result section “Viruses target highly-connected and central host proteins involved in critical cellular functions”, paragraph 2, page 9).

4. Figure 2 would benefit from the inclusion of the information of which compartment the authors would assign each protein, based on the imaging. The authors could also compare their MS-microscopy results (Supplementary Figure 12) with the actual microscopy. Do all proteins agree?

This was a good suggestion. We have included an additional display item into the revised version (**Dataset EV7**). We have compared results with previous publications (Gordon, Hiatt et al. 2020, Zhang, Cruz-cosme et al. 2020, Lee, Huang et al. 2021), no obvious difference was observed by light microscopy of the subcellular localization of single ORF expressing in several transformed cell lines (Dataset EV7). MS-Microscopy can assign the protein in specific subcellular cytosol organelles/structures, therefore, sub-cellular localizations of viral ORFs were presenting in more specific organelles/structures by MS-Microscopy. This data has now been added to the manuscript (**Result section “Subcellular localization of viral bait proteins”, page 7**).

5. The section "Functional characteristics of hub proteins involving in critical signaling pathways" reads mostly like a review of the literature and I feel that it falls short of putting the results in context. While figure 4 points out the frequency at which these proteins were preys in the experiments from the authors, they do not link to specific viral proteins. The authors should consider rewriting this section to link specific viral proteins to the host processes highlighted.

We thank the Reviewer for this suggestion. We have rewritten the description of the result to highlight specific proteins involved in the host processes.

Endosomal entry mechanisms provide many advantages to the virus by allowing SARS-CoV-2 to efficiently spread while avoiding host immunological surveillance (Bayati, Kumar et al. 2020). We detected the viral ORFs interact with 47 host proteins on the endocytosis pathway. Ten viral ORFs had more than 10 interactions with the proteins of this pathway, including M (27 interactions), Orf3a (26), ORF7a (19), NSP6 (18), S (15), ORF3b (12), E (11), NSP7 (11), ORF10 (11), and NSP10 (10) (Fig 4C).

Viral protein intake (E, M, NSP3, 4, 6, 9, 13, 14, ORF3a, 7a) is regulated by clathrin/AP2 complex-mediated endocytosis with CDC42 (Fig 4C) (Swaine and Dittmar 2015, Bayati, Kumar et al. 2020), after which viral particles (NSP5, 6, 7, 16, ORF3a, 7a) end up in RAB5-containing early endosomes (Fig 4C)...

On the protein processing in the endoplasmic reticulum pathway the viral proteins interact with 41 of the pathway components, and nine viral ORFs have more than ten interactions with them. The M proteins interact with 27 of the pathway proteins, ORF7a (23), ORF3a (21), ORF10 (21), S (20) ORF8 (17), NSP6 (15), E (12), and ORF6 (11) (Fig 4D)...

6. The methods section lacks most aspects of how data analysis was performed. A few examples: how the MS-microscopy was carried out?, how the correlations in Figure 3 were performed?. It would be important to go through all figures/analyses and include these details in the methods.

We have carefully examined the manuscript for descriptions of methodology and improved the description of the methodology. Every Method section and/or corresponding figure legends have updated into more details.

7. It would be good if supplementary tables 1 and 2 were more harmonized. For example, Supplementary table 2 lacks the information if host proteins were tagged in the C- or N-terminus.

The tag information of host bait proteins was presented in the original Dataset, but we understand that some of the Datasets were apparently lacking relevant details and descriptions, and needed further clarification. We have updated all the Datasets and added details where needed. The revised Dataset EV1 should now include all the information of the bait protein.

Thank you for sending us your revised manuscript. We have now heard back from the two reviewers who were asked to evaluate your revised study. As you will see below, the reviewers think that the study has improved as a result of the performed revisions. However, reviewer #2 raises a few remaining concerns, which we would ask you to address in a minor revision.

We would also ask you to address few editorial issues listed below.

REFeree REPORTS

Reviewer #2:

It is true the authors have produced a lot of data which will be useful for the scientific community. There are a few remaining concerns I have with the manuscript that I have outlined below.

- The author's response to "Result section "Viruses target highly-connected and central host proteins involved in critical cellular functions", paragraph 2, page 9" may be correct but left me confused. Originally I was expecting the authors to prove that the preys of viral bait proteins were enriched as "hub" proteins (high degree) in other host-host interaction networks (IntAct, STRING, PathwayCommons, etc). The Appendix Figure S10 also left me confused. For instance, the authors should clarify what they mean by: "Adjusted protein frequency of occurrence". If the authors could clarify this analysis in the text and in the figure, it would be much easier to understand by future readers. The point is to prove that viral bait proteins target host proteins that are more "interconnected" with other host proteins or more "essential to" the host signaling network.

- With regards to the drug studies. I applaud the authors for doing this work, as it is not easy to do. Yet, I am a bit confused as to why for many of them percent inhibition starts around 50%. Could this be a problem with their no-drug-treated virus infected control to which everything is compared? I would be surprised if 0.5nM of drug had antiviral effects.

Furthermore, the IC50 and EC50 values should reflect the concentration at which the dose response curve crosses the 50% line, and not solely based on the model. If the dose response curve is on a 0-100% scale, then the IC50 and EC50 should be defined in relation to that.

I might say Baicalein has antiviral properties but I am not very convinced by the data from Methotrexate, Guadecitabine, PX-478, Mizoribine, and BMS-863233 as they seem to hover around 50%, which again might be affected by the positive control (virus but no drug).

Reviewer #3:

The authors have addressed all of my comments and performed extensive validation of some of their newly identified PPIs, as well as potential antiviral drugs identified by the virtual screening. I also feel that they have addressed the comments from reviewer 2. Therefore, I congratulate the authors on the extensive work presented and recommend that this should be published.

Point-by-point response to the referees

Reviewer #2:

It is true the authors have produced a lot of data which will be useful for the scientific community. There are a few remaining concerns I have with the manuscript that I have outlined below.

We thank the Reviewer for her/his positive view of our revised manuscript.

- The author's response to "Result section "Viruses target highly-connected and central host proteins involved in critical cellular functions", paragraph 2, page 9" may be correct but left me confused. Originally I was expecting the authors to prove that the preys of viral bait proteins were enriched as "hub" proteins (high degree) in other host-host interaction networks (IntAct, STRING, PathwayCommons, etc). The Appendix Figure S10 also left me confused. For instance, the authors should clarify what they mean by: "Adjusted protein frequency of occurrence". If the authors could clarify this analysis in the text and in the figure, it would be much easier to understand by future readers. The point is to prove that viral bait proteins target host proteins that are more "interconnected" with other host proteins or more "essential to" the host signaling network.

We thank the Reviewer for the advice and apologize for causing unnecessary confusion. We have checked the connection of these 693 proteins in human interactome database and the result indicates most of these proteins indeed have more inter-connections than other proteins in the human interactome (Appendix Fig S10A). This result were included to the result section "Viruses target highly-connected and central host proteins involved in critical cellular functions" (paragraph 2, page 9). We have updated the method, text and appendix figure and legend accordingly to avoid possible misunderstanding.

- With regards to the drug studies. I applaud the authors for doing this work, as it is not easy to do. Yet, I am a bit confused as to why for many of them percent inhibition starts around 50%. Could this be a problem with their no-drug-treated virus infected control to which everything is compared? I would be surprised if 0.5nM of drug had antiviral effects.

Furthermore, the IC50 and EC50 values should reflect the concentration at which the dose response curve crosses the 50% line, and not solely based on the model. If the dose response curve is on a 0-100% scale, then the IC50 and EC50 should be defined in relation to that.

I might say Baicalein has antiviral properties but I am not very convinced by the data from Methotrexate, Guadecitabine, PX-478, Mizoribine, and BMS-863233 as they seem to hover around 50%, which again might be affected by the positive control (virus but no drug).

We apologize for any unnecessary confusion. We have included the control with 0 nM concentration (virus-infected cells+DMSO) as the starting point and updated all graphs to improve the curve fitting of drug response. We agree with the Reviewer that the original curve fitting algorithm was too stringent for measuring the effects of drugs on virus infection. Therefore, instead

of using drug sensitivity scoring (DSS) generated by automated pipeline (Breeze), which are designed for dose-response curve fitting of effect of oncological compounds in cancer cells, we re-calculated the IC50 and the area under the dose-response curve (AUC) based on updated graphs for these 10 drugs. We have accordingly modified the main text, Fig.6, Appendix Fig.S11, DatasetEV13, and relevant method parts.

As we were limited in time and resources, the antiviral effect of drugs needs further validation in other physiologically relevant cell lines and animal models. However, Baicalein, Methotrexate, Guadecitabine, PX-478, Mizoribine, and BMS-863233 all showed some antiviral effects using both (earlier and current) analysis approaches. Moreover, except Guadecitabine, all the rest 5 drugs have been suggested the antiviral effect on SARS-CoV-2 by other recent studies (**Dataset EV12**).

Reviewer #3:

The authors have addressed all of my comments and performed extensive validation of some of their newly identified PPIs, as well as potential antiviral drugs identified by the virtual screening. I also feel that they have addressed the comments from reviewer 2. Therefore, I congratulate the authors on the extensive work presented and recommend that this should be published.

We thank the Reviewer again for her/his very constructive suggestions for revising the manuscript.

Thank you again for sending us your revised manuscript. We are now satisfied with the modifications made and I am pleased to inform you that your paper has been accepted for publication.

Corresponding Author Name: Markku Varjosalo

Manuscript Number: MSB-2021-10396